# Predictors of human-infective RNA virus discovery in the United States, China, and Africa, an ecological study

**Feifei Zhang[1]\*, Margo Chase-Topping[1,2], Chuan-Guo Guo[3], Mark EJ Woolhouse[1]**

[1]Usher Institute, University of Edinburgh, Edinburgh, United Kingdom; [2]Roslin Institute and Royal (Dick) School of Veterinary Studies, University of Edinburgh, Edinburgh, United Kingdom; [3]Department of Medicine, Li Ka Shing Faculty of Medicine, University of Hong Kong, Hong Kong, China

## Abstract

**Background:** The variation in the pathogen type as well as the spatial heterogeneity of predictors make the generality of any associations with pathogen discovery debatable. Our previous work confirmed that the association of a group of predictors differed across different types of RNA viruses, yet there have been no previous comparisons of the specific predictors for RNA virus discovery in different regions. The aim of the current study was to close the gap by investigating whether predictors of discovery rates within three regions—the United States, China, and Africa—differ from one another and from those at the global level.

**Methods:** Based on a comprehensive list of human-infective RNA viruses, we collated published data on first discovery of each species in each region. We used a Poisson boosted regression tree (BRT) model to examine the relationship between virus discovery and 33 predictors representing climate, socio-economics, land use, and biodiversity across each region separately. The discovery probability in three regions in 2010–2019 was mapped using the fitted models and historical predictors.

**Results:** The numbers of human-infective virus species discovered in the United States, China, and Africa up to 2019 were 95, 80, and 107 respectively, with China lagging behind the other two regions. In each region, discoveries were clustered in hotspots. BRT modelling suggested that in all three regions RNA virus discovery was better predicted by land use and socio-economic variables than climatic variables and biodiversity, although the relative importance of these predictors varied by region. Map of virus discovery probability in 2010–2019 indicated several new hotspots outside historical high-risk areas. Most new virus species since 2010 in each region (6/6 in the United States, 19/19 in China, 12/19 in Africa) were discovered in high-risk areas as predicted by our model.

**Conclusions:** The drivers of spatiotemporal variation in virus discovery rates vary in different regions of the world. Within regions virus discovery is driven mainly by land-use and socio-economic variables; climate and biodiversity variables are consistently less important predictors than at a global scale. Potential new discovery hotspots in 2010–2019 are identified. Results from the study could guide active surveillance for new human-infective viruses in local high-risk areas.

**Funding:** FFZ is funded by the Darwin Trust of Edinburgh (https://darwintrust.bio.ed.ac.uk/). MEJW has received funding from the European Union's Horizon 2020 research and innovation programme under grant agreement No. 874735 (VEO) (https://www.veo-europe.eu/).

\*For correspondence:
Feifei.Zhang@ed.ac.uk

**Competing interest:** The authors declare that no competing interests exist.

## Editor's evaluation

This study will be of interest to readers in the field of virus discovery. This study attempts to identify predictors of human-infective RNA virus discovery and predict high risk areas in a recent period in the United States, China, and Africa using an ecological modeling framework. The study has potential to inform future discovery efforts for human-infective viruses.

## Introduction

RNA viruses are the primary cause for emerging infectious diseases with epidemic potential, given that they have a high rate of evolution and high capacity to adapt to new hosts (*Woolhouse et al., 2016*). In recent decades, infectious diseases caused by severe acute respiratory syndrome coronavirus (SARS-CoV), Middle East respiratory syndrome coronavirus (MERS-CoV), Bundibugyo Ebola virus and SARS-CoV-2 present major threats to the health and welfare of humans (*Albariño et al., 2013*; *Ksiazek et al., 2003*; *Mackay and Arden, 2015*; *World Health Organisation, 2020*). Detection of formerly unknown human-infective RNA viruses in the earliest stage after the emergence are essential for controlling the infections they cause. Measures to implement early detection include not only advanced diagnostic techniques (*Lipkin and Firth, 2013*), but more importantly the idea where to look for them (so-called hotspots) (*Morse, 2012*).

Socio-economic, environmental, and ecological factors related to both virus natural history and research effort have been found to affect the discovery of emerging RNA viruses (*Jones et al., 2008*; *Morse, 2012*; *Rosenberg, 2015*; *Zhang et al., 2020*). However, these factors are highly spatially heterogeneous, making the generality of any associations with discovery debatable. For example, the United States, China, and Africa have experienced different rates of socio-economic, environmental, and ecological changes in the last one hundred years. The United States has always had better resources to discover new viruses. For example, the Rockefeller Foundation—a U.S. foundation—supported the discovery of 23 arboviruses in Latin America, Africa, and India in 1951–1969 (*Rosenberg et al., 2013*). China has seen urban land coverage more than double and GDP per capita increase by seven times since the 1980s (*Ritchie, 2018*; *Roser, 2013*). Nine out of 223 human-infective RNA viruses have been originally discovered in China, and all were discovered after 1982 (*Zhang et al., 2020*). In contrast, effective surveillance is challenging in less developed regions such as large parts of Africa given resource constraints (*Petti et al., 2006*).

There have been no previous comparisons of the specific predictors for RNA virus discovery in different regions. In this study, we applied a similar methodology from our previous study of global patterns of discovery of human-infective RNA viruses (*Zhang et al., 2020*) to investigate whether predictors of discovery rates within three regions—the United States, China, and Africa—differ from one another and from those at the global level, using three new virus discovery data sets. We also mapped discovery probability in three regions in 2010–2019 using the fitted models and historical predictors. According to findings from our previous study (*Zhang et al., 2020*), the main predictors for virus discovery at the global scale were GDP-related. This suggests that the patterns of virus discovery we have identified may have been largely driven by research effort rather than the underlying biology. In this study, by focusing on more restricted and homogenous regions where the research effort is less variable, we expected to identify predictors more associated with virus biology.

## Materials and methods

### Data sets of human-infective RNA viruses in three regions

We performed an ecological study, and the subject of interest is each human-infective RNA virus species. With reference to a full list of human-infective RNA virus species (*Zhang et al., 2020*), we geocoded the first report of each in humans in the United States, China, and Africa separately. The latest version as of 31 December 2019 included 223 species (*Appendix 1—table 1*), with *Human torovirus* abolished and a new species—*Heartland banyangvirus*—added by International Committee on Taxonomy of Viruses (ICTV) in 2018 (*International Committee on Taxonomy of Viruses, 2018*). Data used in this study were not subsets of our previous global analysis; information on discovery locations and discovery dates for each virus species was re-collated for each specific geographical region.

We followed the same search terms, databases searched, and inclusion or exclusion criteria as our global data set for data collection (*Woolhouse and Brierley, 2018*). In each region, we established whether or not each virus species has been discovered in humans according to peer-reviewed literature. Reference databases included PubMed, Web of Science, Google Scholar, and Scopus. Two Chinese databases [i.e. China National Knowledge Infrastructure (CNKI) and Wanfang Data] were also searched when collecting data for China. Reference lists of relevant studies and reviews were also checked manually to find potential earlier discovery papers. The following key words were used for the retrieval: virus full name or abbreviations or virus synonyms; and human* or person* or case* or patient* or worker* or infection* or disease* or outbreak* or epidemic*; and region name (Chin* or Taiwan or Hong Kong or Macau; United States or US or USA or America*; Africa* or all African country names). Virus synonyms and abbreviations include early names used in the discovery paper and all subtypes provided by the ICTV online report (*International Committee on Taxonomy of Viruses, 2018* ). Evidence which met the following criteria from peer-reviewed literatures were included: (a) Diagnostic methods for RNA virus infection in humans were clearly described, through either viral isolation or serological methods; (b) Specific virus species name or subtypes falling under that species were clearly provided; (c) Both natural infection and iatrogenic or occupational infections were accepted. Evidence which met the following criteria were excluded: (a) Uncertain species due to cross-reactivity with related viruses; (b) Diagnostic methods for virus infection were not specified; (c) Description of clinical symptoms or pathogenicity were not considered as human infection of one certain virus species; (d) Report of '[virus name]-like' or 'potential [virus name] infections'; (e) Intentional infections including experimental inoculation or vitro infections; (f) Non-peer-reviewed literature, including media reports, thesis, or unpublished data. Literature selection was performed by two individuals independently and discrepancies were resolved by discussion with a third individual.

We defined discovery location as where the initial human was exposed to/infected with the virus, as suggested in the first report of human infections from peer-reviewed literature. All locations were geolocated as precisely as possible using methods from our previous paper (*Zhang et al., 2020*). For each region, a polygon was created for those locations at administrative level 3 (county for the United States; city for China; for Africa, it varies between different countries) and above. Details of data types for virus discovery database in three regions was summarised in *Appendix 1—table 2*. Although the majority of discovery locations in the United States and Africa involved point data and in China the majority involved polygon data at province level, the average number of grid cells per virus in three regions were similar. A bootstrap resampling procedure was developed for polygon data covering more than one grid cell (details below). Discovery date of human infection was defined as the publication year in the scientific literature.

## Spatial covariates

As for our global analysis (*Zhang et al., 2020*), a suite of global gridded climatic, socio-economic, land use, and biodiversity variables (n=33) postulated to affect the spatial distribution of RNA virus discovery were compiled, each at a resolution of 0.5°/30" (except university count having a resolution at country level for Africa and at state/province level for the United States and China). Of these, GDP, GDP growth, and university were included to adjust for discovery effort as they could partially explain the infrastructure and technology that are available for virus research (*Zhang et al., 2020*). We reviewed and tested previous strategies researchers have used to adjust for discovery bias, including frequency of the country listed as the address for authors in scientific papers and frequency of publications for each pathogen from scientific databases (*Jones et al., 2008*; *Olival et al., 2017*) but the results were not encouraging as the frequency of published papers from virus-related scientific journals is weakly linked to the published count of novel human-infective RNA virus (Appendix2, *Appendix 3—figure 1*).

Data for the United States, China, and Africa were extracted by restricting the coordinates within each region. The definition, original resolution, and source of each variable were the same as our previous paper (*Zhang et al., 2020*). All predictors were aggregated from their original spatial resolution to 1°×1° resolution; data for climatic variables, population, GDP, and land use data without full

temporal coverage were extrapolated back to 1901; both following methods from our previous paper (*Zhang et al., 2020*).

## Boosted regression trees modelling

We used a Poisson boosted regression trees (BRT) model to examine the relationship between discovery of RNA virus and 33 predictors for each 1° resolution of grid cell across each region separately, following codes from our previous study (*Zhang et al., 2020*) and one previous paper (*Allen et al., 2017*). As a tree-based machine learning method, the BRT model can automatically capture complex relationships and interactions between variables, and also can well account for spatial autocorrelation within the data (*Crase et al., 2012*). We compared Moran's I values of the raw virus data and the model residuals to estimate the ability of the BRT model to account for spatial autocorrelation (*Cliff and Ord, 1981*). In order to minimise the effect of spatial uncertainty of virus discovery data, we performed 1000 times bootstrap resampling for those discovery locations reported as polygons. We assumed each grid cell in the polygon has the equal chance to be selected, and for each virus record we selected one grid cell randomly from the polygon for each subsample. A ratio of 1:2 for presence to absence constituted each subsample, that is, for each grid cell with virus discovery, two grid cells with no discovery were randomly selected from 'virus discovery free' areas at all time points within the region. Take the United States as an example, each subsample included 95 grid cells with virus discovery and 190 with no virus discovery. We then matched the virus data with all predictors by geographical coordinates and decade (using the nearest decade for time-varying predictors). We assumed that the virus count in any given grid cell in each decade followed a Poisson distribution, and we calculated the virus discovery count in each grid cell by decade as the response variable.We also performed further sensitivity analyses by (i) matching virus discovery data and time-varying covariate data by year and (ii) testing for lag effects by matching virus discovery at year t and predictors at t-1 to t-5 year (Appendix4).

All BRT models were fitted in R v. 3.6.3, using packages dismo and gbm. BRT models require the user to balance three parameters including tree complexity, learning rate, and bag fraction. Tree complexity reflects the order of interaction in a tree; learning rate shrinks the contribution of each tree to the growing model; bag fraction specifies the proportion of data drawn from the full training data at each step. We set these parameters as recommended from *Elith et al., 2008*, and make sure each resampling model contained at least 1000 trees. BRT models identified the final optimal number of trees in each model using a 10-fold cross validation stagewise function (*Elith et al., 2008*). The three parameter values of the optimal model as well as the mean optimal number of trees across 1000 replicate models for all three regions were summarised in *Appendix 1—table 3*.

By fitting 1000 replicate BRT models, the relative contribution plots and partial dependence plots with 95% quantiles were plotted. We defined variables with a relative contribution greater than the mean (3.03%) as influential predictors in all three regions (*Shearer et al., 2018*). The partial dependence plots depict the influence of each variable on the response while controlling for the average effects of all the other variables in the model. The map of virus discovery probability across each region in 2010–2019 was derived from the means of the predictions of 1000 replicate models, using values of the 33 predictors in 2015. In order to show discovery hotspots, we converted the prediction map of virus count to a map of probability.

Two statistics were calculated to evaluate the model's predictive performance: (a) the deviance of the bootstrap model (*Elith et al., 2008*), (b) intraclass correlation coefficient (ICC) calculated from 50 rounds of 10-fold cross-validation, by following methods from our previous paper (*Zhang et al., 2020*). For the 10-fold cross-validation, we selected 50 data sets randomly from the 1000 bootstrapped subsamples. We took the first data set and partitioned into 10 subsets. For each round of 10-fold cross-validation, the unique combinations of nine subsets constituted the training sets and were used to fit models, and the remaining one was used as a test set to evaluate the predictive performance of the model. We repeated the same process as above for the remaining 49 data sets. One intraclass correlation coefficient (ICC) was calculated from each round of validation and the median with 95% quantiles across all 50 rounds was calculated. The ICC varies between 0 and 1, with an ICC of less than 0.40 representing a poor model, 0.40–0.59 representing a fair model, 0.60–0.74 representing a good model, and 0.75–1 representing an excellent model (*Cicchetti, 1994*).

Exploratory subgroup analyses distinguishing viruses firstly discovered in regions and those that had been discovered elsewhere in the world were performed. We used the same BRT modelling

approach as we described above, and relative contribution of each predictor was calculated for each subgroup. We were unable to perform subgroup analysis for China because only nine human-infective RNA viruses have been firstly discovered in it, and the BRT model cannot be fitted to a sample as small as 9.

R software, version 3.6.3 (R Foundation for Statistical Computing, Vienna, Austria) was used for all statistical analyses. All maps were visualised by using ArcGIS Desktop 10.5.1 (Environmental Systems Research Institute).

## Results

The numbers of human-infective virus species discovered in the United States, China, and Africa up to October 2019 were 95, 80, and 107, respectively (*Appendix 1—table 1*). Most first discoveries have been in eastern United States (especially in areas around Maryland, Washington, D.C., and New York), eastern China (developed cities including Beijing, Hong Kong, Shanghai, and Guangzhou), and southern and central Africa (Pretoria and Johannesburg, South Africa; Borno State and Ibadan, Nigeria) (*Figure 1*). A total of 60 virus species were previously reported in all three regions, and 27, 12, 37 species were only found in the United States, China, and Africa, respectively (*Figure 2*). In all three regions, smaller proportions of viruses were vector-borne [United States: 23.2% (22/95); China: 21.3% (17/80); Africa: 27.1% (29/107)] and strictly zoonotic [United States: 30.5% (29/95); China: 16.3% (13/80); Africa: 33.6% (36/107)], compared to large proportions for both virus types at the global scale [vector-borne: 41.7% (93/223) and strictly zoonotic: 58.7% (131/223)] (*Figure 2*). The 60 shared species were also disproportionally vector-borne [11.7% (7/60)] and strictly zoonotic [7% (4/60), *Figure 2*].

The discovery curves for the United States and Africa have seen a broadly similar pattern, with China lagging behind these two regions (*Figure 3*). The median time lag between the original discovery year of each virus in the world and the discovery year of each virus in each region was 0 [interquartile range (IQR): 2.5], 12 (IQR: 29.5), and 2 (IQR: 10.5) years in the United States, China, and Africa, respectively (*Appendix 3—figure 2*). In China, the time lag was noticeably shorter for viruses discovered after 1975 [before 1975: a median lag of 30.5 (IQR: 30.5) years; after 1975: 2.5 (IQR: 7) years, p value of Wilcoxon rank sum test < 0.001].

In the United States, six variables including three predictors related to land use [urbanized land: relative contribution of 35.8%, urbanization of cropland (i.e. the percentage of land area change from cropland to urban land): 8.0%, growth of urbanized land: 4.1%], two socio-economic variables (GDP growth: 10.0%; GDP: 5.7%), and one climatic variable (diurnal temperature change: 4.9%) were identified as important predictors for discriminating between locations with and without virus discovery (*Figure 4A*). The partial dependence plots shown in *Appendix 3—figure 3* suggested non-linear relationships between the probability of virus discovery and most predictors. All important predictors presented a positive trend over narrow ranges at lower values.

In China, twelve variables including four socio-economic variables (GDP: 12.7%, university count: 7.5%, GDP growth: 4.6%, population growth: 4.4%), five predictors involving land use [pasture: 8.3%, urbanized land: 8.1%, vegetation: 5.8%, cropland: 5.3%, urbanization of secondary land (the percentage of land area change from secondary land to urban land; secondary land is natural vegetation that is recovering from previous human disturbance): 3.3%], and three climatic variables (maximum precipitation: 4.5%, precipitation change: 3.8%, diurnal temperature range: 3.3%) were identified as important predictors for discriminating between locations with and without virus discovery (*Figure 4B*). GDP, urbanized land, university count, vegetation, GDP growth, maximum precipitation, population growth, and urbanization of secondary land presented a positive trend over narrow ranges at lower levels; pasture, cropland, precipitation change, and diurnal temperature range had non-monotonic/negative impacts, with highest risks at lower values (*Appendix 3—figure 4*).

In Africa, ten variables including two socio-economic variables (GDP growth: 21.2%, GDP: 13.0%), seven predictors related to land use (urbanized land: 9.4%, growth of cropland area: 5.6%, urbanization of cropland: 5.5%, growth of urbanized land: 5.1%, urbanization of pasture: 3.8%, vegetation, 3.7%, cropland: 3.2%), and one biodiversity variable (mammal species richness: 3.1%) were identified as important predictors for discriminating between locations with and without virus discovery (*Figure 4C*). All important predictors presented a positive trend over narrow ranges at lower positive values, except mammal species over a large range (*Appendix 3—figure 5*).

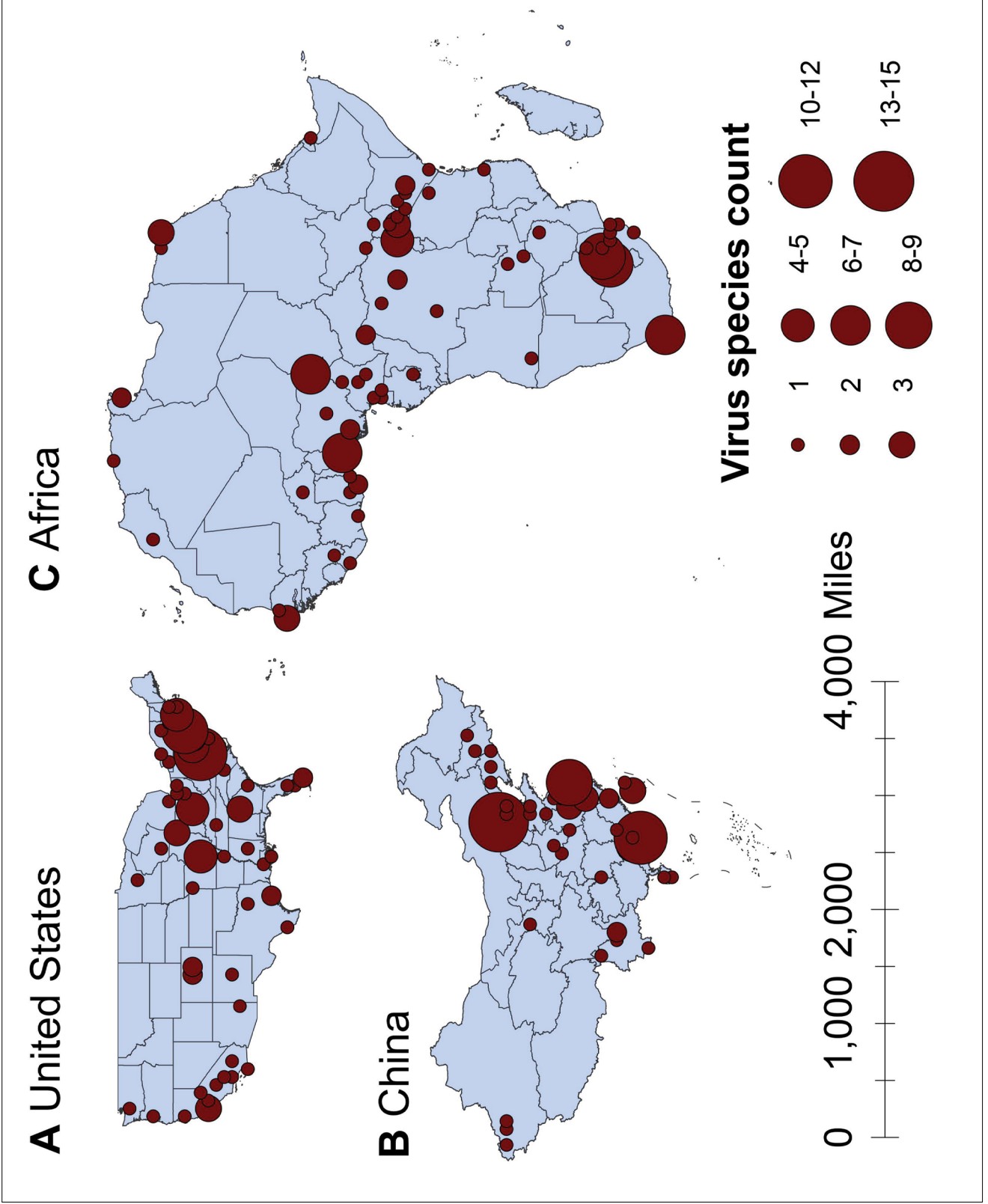

**Figure 1.** Spatial distribution of human-infective RNA virus discovery in three regions, 1901–2019. (**A**) United States. (**B**) China. (**C**) Africa. Red dots represent discovery points or centroids of polygons, with the size representing the cumulative virus species count.

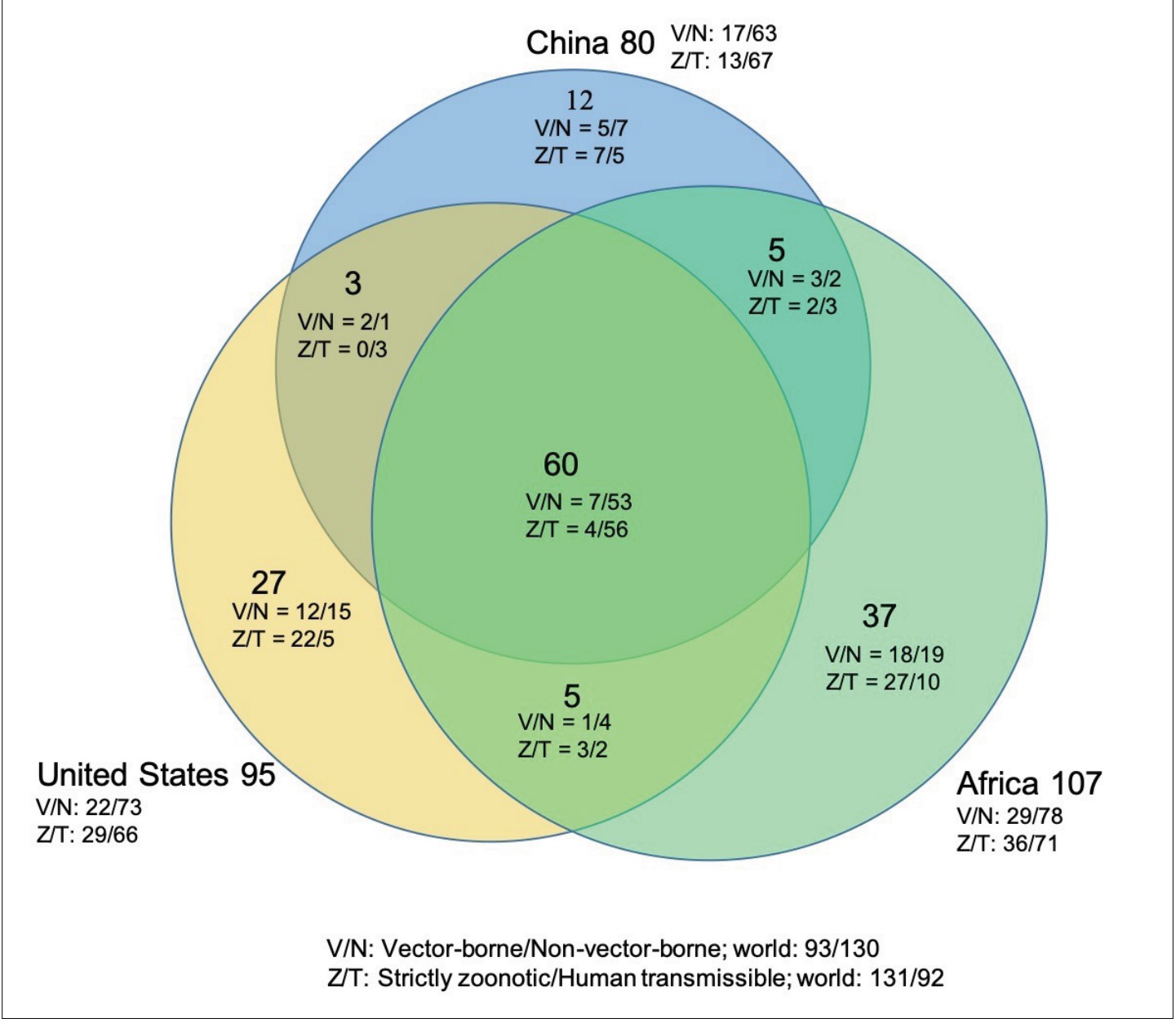

**Figure 2.** Shared human-infective RNA virus species count in three regions. Under/By the species count the ratios of vector-borne (**V**) to non-vector-borne (**N**) viruses and strictly zoonotic (**Z**) to human transmissible (**T**) viruses were shown.

Our BRT models reduced Moran's I value below 0.15 in all three regions (*Appendix 3—figure 6*), suggesting that BRT models with 33 predictors have adequately accounted for spatial autocorrelations in the raw virus data in all three regions. The model validation statistics for each region are shown in *Appendix 1—table 4*. Combining these measures, our BRT model predictions range from fair to good (*Cicchetti, 1994*). In our sensitivity analyses based on data matched by year (*Appendix 3—figure 7*) and 1–5 year lag (results of 1 year lag shown in *Appendix 3—figure 8*), though there were several changes of relative contribution, the top predictors were broadly consistent with our main model based on data matched by decade (*Figure 4*).

In comparison with the whole world, human-infective RNA virus discovery was more associated with land use and socio-economic variables than climatic variables and biodiversity in all three regions (*Figure 5*). The comparison of four groups of predictors between three regions showed that: the greatest contribution of climatic variables to the discovery of human-infective RNA viruses was in

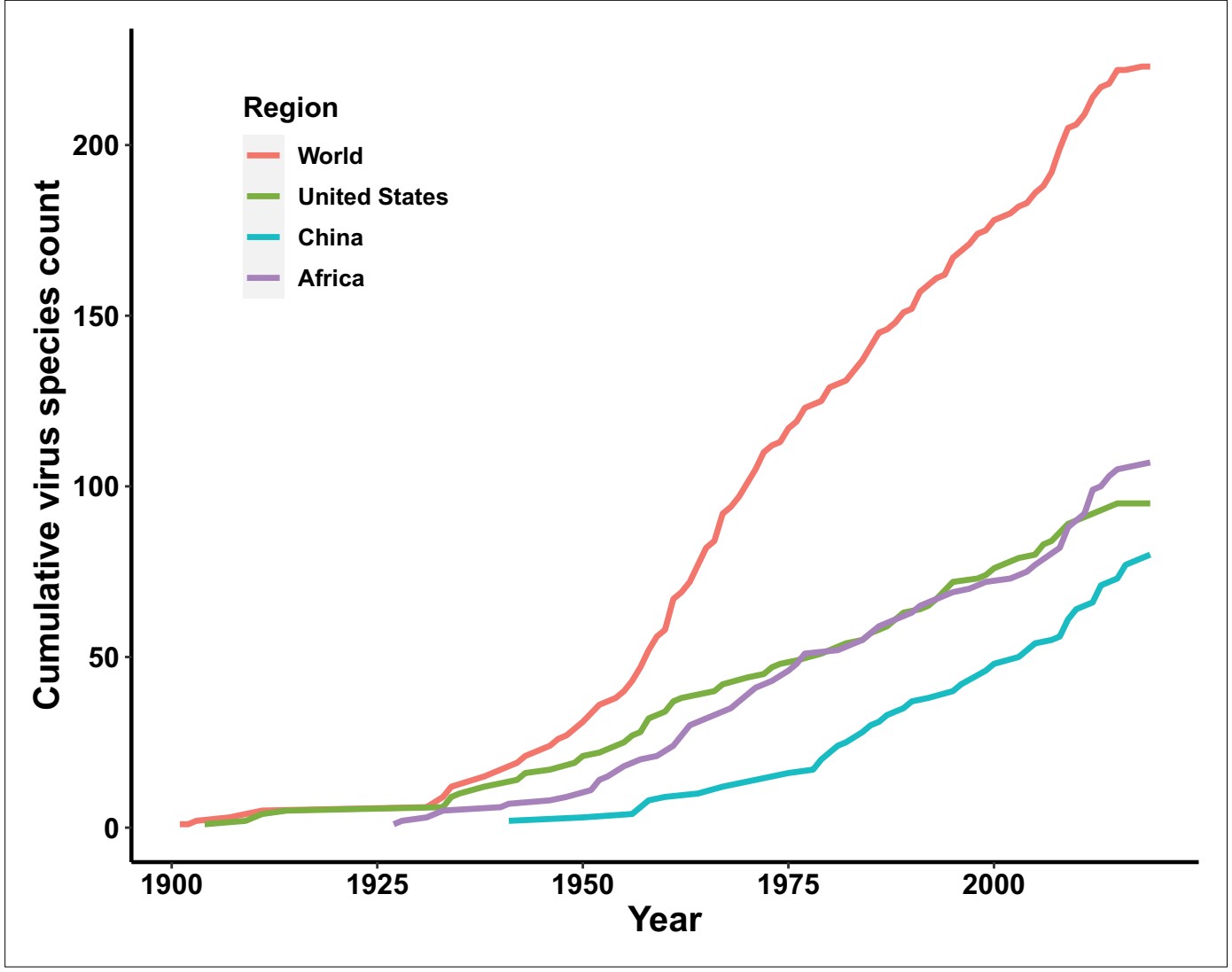

**Figure 3.** Discovery curve of human-infective RNA virus species in three regions and the world.

China; the greatest contribution of land use was in the United States; the greatest contribution of socio-economic variables and biodiversity was in Africa and least in the United States.

We mapped human-infective RNA virus discovery probability in 2010–2019 for the three regions, based on the fitted BRT models and values of all 33 predictors in 2015 (*Appendix 3—figure 9* to *Appendix 3—figure 11*). Outside contemporary risk areas where human-infective RNA viruses were previously discovered in the United States (*Figure 1A*), we predicted high probabilities of virus discovery across southern Michigan, central-Northern Carolina, central Oklahoma, southern Nevada, and north-eastern Utah (*Figure 6A*). Outside contemporary risk areas where human-infective RNA viruses were previously discovered in China (*Figure 1B*), we predicted high probabilities of virus discovery across other eastern China area as well as two western areas including south-central Shaanxi and north-eastern Sichuan (*Figure 6B*). Outside contemporary risk areas where human-infective RNA viruses were previously discovered in Africa (*Figure 1C*), we predicted high probabilities of virus discovery across northern Morocco, northern Algeria, northern Libya, south-eastern Sudan, central Ethiopia and western Democratic Republic of the Congo (*Figure 6C*). Most new virus species since 2010 in each region (6/6 in the United States, 19/19 in China, 12/19 in Africa) were discovered in high-risk areas (85% percentiles of predicted probability across each region) as predicted by our model. Of all the 37 (United States: 6; China: 19; Africa: 12) viruses discovered in high-risk areas in 2010–2019,

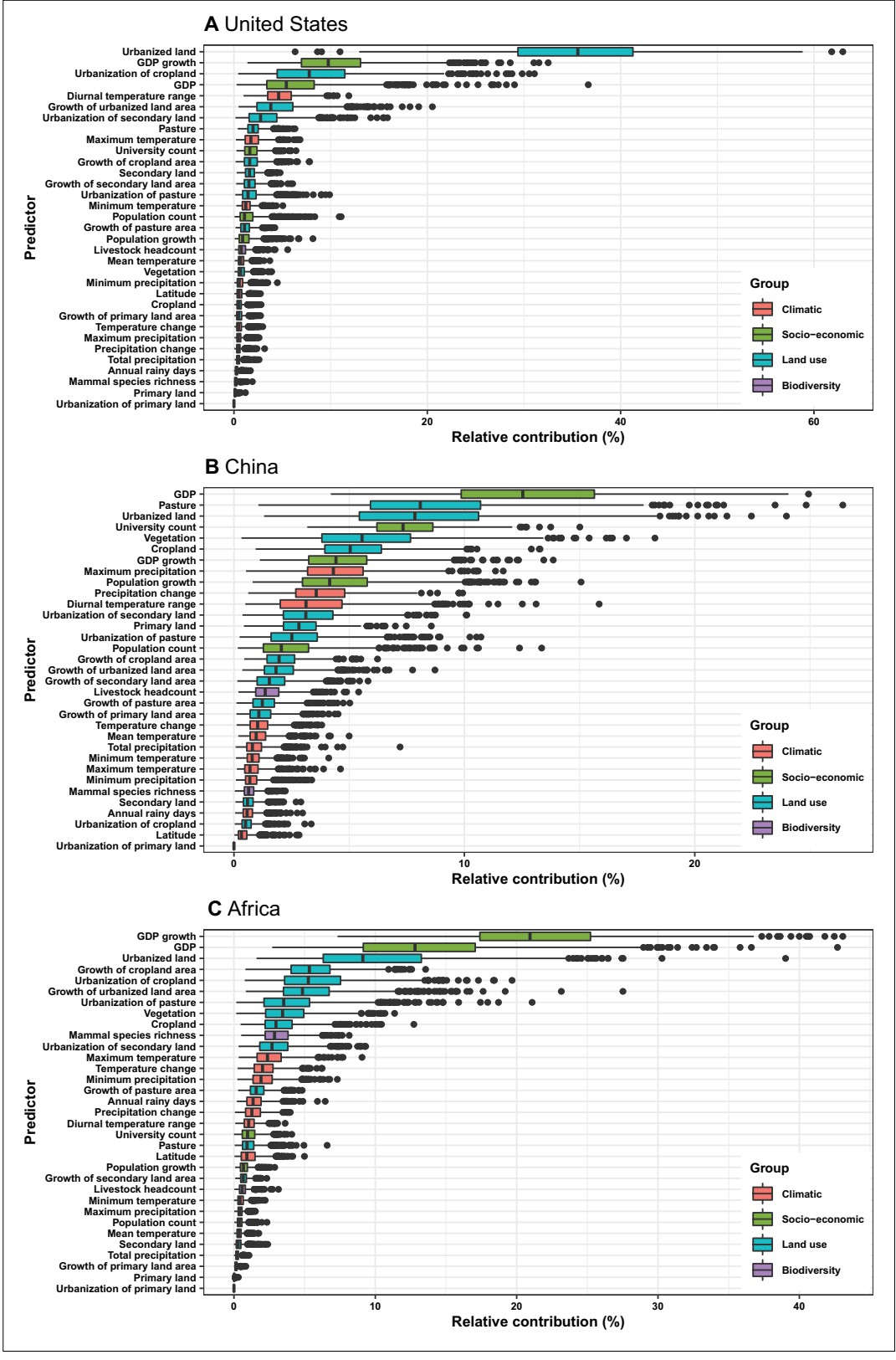

**Figure 4.** Relative contribution of predictors to human-infective RNA virus discovery in three regions. (**A**) United States. (**B**) China. (**C**) Africa. The boxplots show the median (black bar) and interquartile range (box) of the relative contribution across 1000 replicate boosted regression tree models, with whiskers indicating minimum and maximum and black dots indicating outliers.

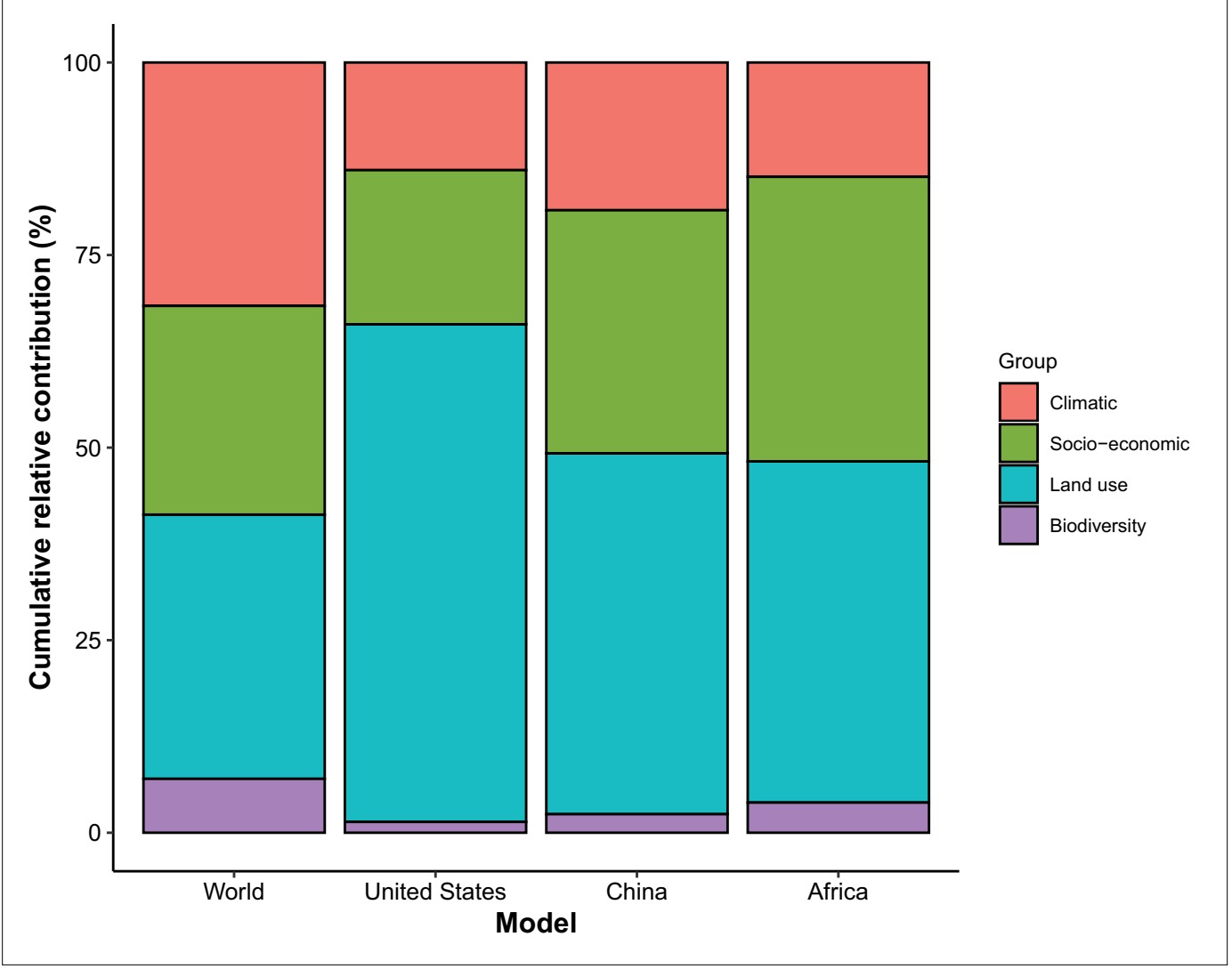

**Figure 5.** Cumulative relative contribution of predictors to human-infective RNA virus discovery by group in each model of different regions. The relative contributions of all explanatory factors sum to 100% in each model, and each colour represents the cumulative relative contribution of all explanatory factors within each group.

13 (United States: 2; China: 7; Africa: 4) viruses were discovered at the potential new hotspots where there have not been any virus discoveries before 2010.

Based on our subgroup analysis distinguishing viruses firstly discovered in regions and those that had been discovered elsewhere in the world, discoveries of human-infective RNA viruses first discovered from either United States or Africa were better predicted by climatic and biodiversity variables, while discoveries of viruses that had been discovered from elsewhere in the world were better predicted by socio-economic variables (*Appendix 3—figure 12*).

## Discussion

To our knowledge, this analysis represents the first investigation of human-infective RNA virus discovery in three large regions of the world which have experienced distinct socio-economic, ecological and environmental changes over the last 100 years. In total, 95 human-infective RNA virus species had been found in the United States; 80 in China; 107 in Africa. The discovery maps of human-infective RNA virus in the three regions indicated areas with historically high discovery counts: eastern and western United States, eastern China, and central and southern Africa. BRT modelling suggested that

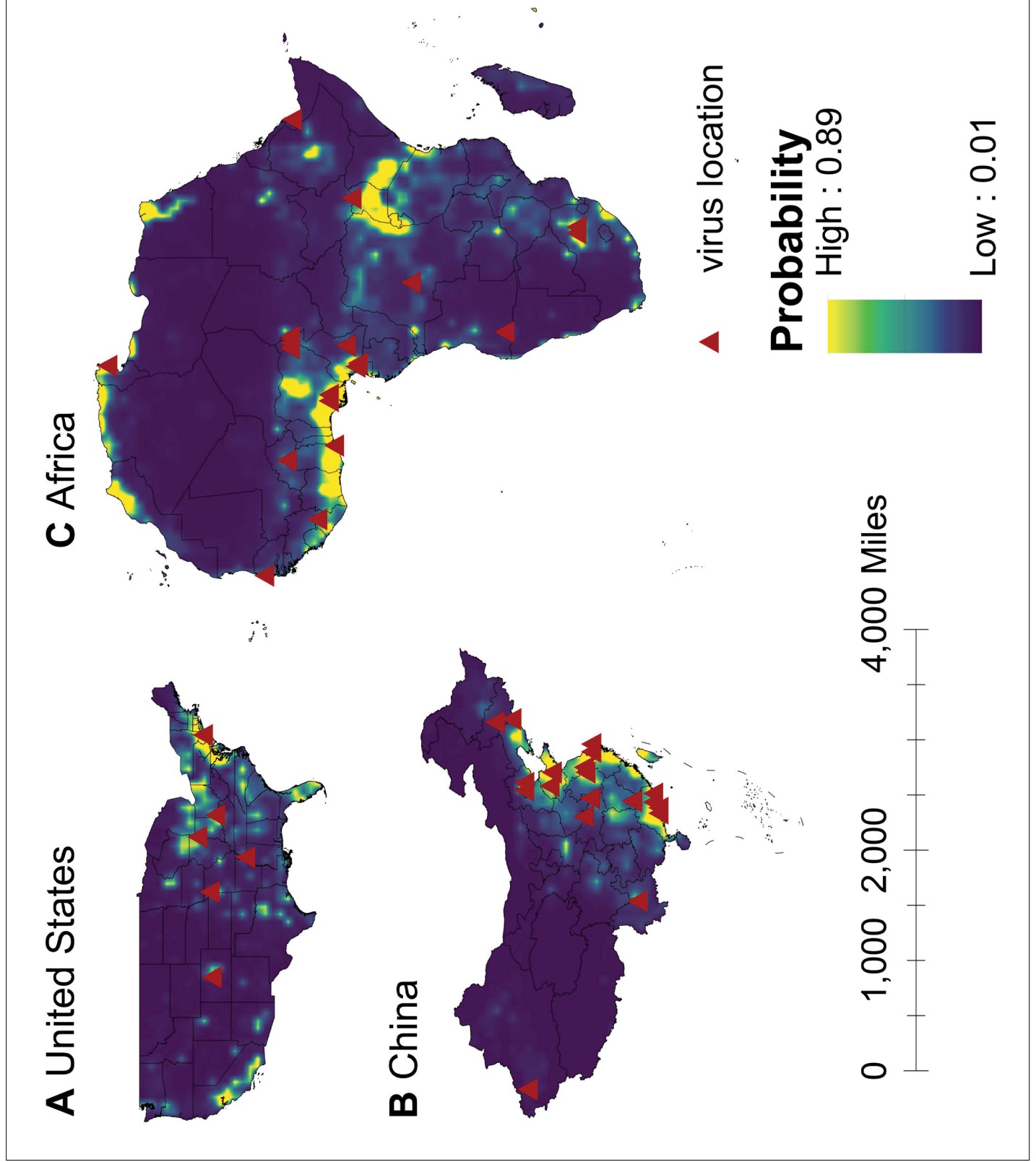

**Figure 6.** Predicted probability of human-infective RNA virus discovery in three regions in 2010–2019. (A) United States. (B) China. (C) Africa. The triangles represented the actual discovery sites from 2010 to 2019, and the background colour represented the predicted discovery probability.

the relative contribution of 33 predictors to human-infective RNA virus discovery varied across three regions, though climatic and biodiversity variables were consistently less important in all three regions than at a global scale. We mapped the probability of human-infective RNA virus discovery in 2010–2019 which would continue to be high in historical hotspots but, in addition, we identified several new hotspots in central-eastern and southwestern United States, eastern and western China, and northern Africa. These results offer a tool for public health practitioners and policymakers to better understand local patterns of virus discovery and to invest efficiently in surveillance systems at the local level.

In recent decades, factors that drive pathogen discovery have been comprehensively studied, e.g., (*Morse, 2012*). In general, evidence has come from three forms of analyses: analysis of single emergence event such as SARS, AIDS, and Ebola (*Parrish et al., 2008*), quantifying the spillover (or host switching/cross-host transmission) risk using traits of both hosts and viruses (*Kreuder Johnson et al., 2015*; *Olival et al., 2017*; *Pulliam and Dushoff, 2009*), and record of first emergence/discovery event in humans globally over time (*Allen et al., 2017*; *Jones et al., 2008*; *Zhang et al., 2020*). Of these, the latter form of analyses have linked the distribution of emerging infectious diseases across the globe to ecological, environmental, and socio-economic factors, predicted the high-risk areas for discovery of emerging zoonoses, and helped identify priority regions for investment in surveillance systems for new human viruses (*Allen et al., 2017*; *Jones et al., 2008*; *Zhang et al., 2020*). In addition to these analyses, our current regional analyses identified more precise hotspots for virus discovery in three large regions of the world. Because zoonotic viruses are responsible for most historical endemics and epidemic diseases, several projects such as the Global Virome project (GVP), the PREDICT project, and the Vietnam Initiative on Zoonotic Infections (VIZIONS) were launched to construct a comprehensive data set of unknown viruses with epidemic potential from specific animals likely to harbour high-risk viruses, humans having a high contacting rate with animals, and animal-human interfaces with high spill-over probability (*Carroll et al., 2018*; *Morse, 2012*; *Rabaa, 2015*). These hotspots analyses indicate priority regions for surveillance for new viruses for these projects.

In all three regions, GDP and/or GDP growth were identified as important predictors for virus discovery. This is consistent with our previous analysis that GDP and GDP growth play a major role in discovering viruses (*Zhang et al., 2020*). In general, sufficient economic, human and material resources, the availability of advanced infrastructure and technology, and greater research capabilities in the relative higher income areas enable the virus discovery (*Rosenberg et al., 2013*). That this effect applied both within one continent and within single countries such as the United States and China suggested that most virus discoveries were likely passive, that is, the viruses were detected when they arrived in a location with the resources to detect them. This is plausible because in all regions in our study, human-transmissible viruses accounted for the larger proportion, and our previous analysis suggested richer areas were more likely to first capture transmissible viruses (e.g. Influenza virus, Rhinovirus, Rabies lyssavirus, Measles morbillivirus, Mumps orthorubulavirus, Rubella virus, and Norwalk virus) capable of spreading to multiple areas (*Zhang et al., 2020*). Temporally, in China the rate of discovery increased after economic growth accelerated in the 1980s (*Figure 3*). We note in publications describing first virus discoveries that most historical virus discoveries in Africa received support from the United States and Europe, and this may explain why Africa saw an increased number of virus discoveries after 1950—30 years earlier than China (*Figure 3*). Notably, in contrast to Africa, university count was found to be associated with virus discovery in China, suggesting virus discovery likely being a significant area of research in Chinese universities. Our model also suggested the overall socio-economic factors contributed less in the United States than other two regions. The possible explanation is that the socio-economic level across the whole United States is relatively high and homogenous.

Predictors other than GDP and university count are likely to be linked to virus natural history. In all three regions, the area of urban land and further urbanization made great contribution to virus discovery. This reinforced previous studies that urbanization was linked to the detection of new human pathogens through the denser urban population, increased human-wildlife contact rate, spill-over of human infection from enzootic cycle, and the contamination of the urban environment with microbial agents (*Hassell et al., 2017*; *Olival et al., 2017*; *Weaver, 2013*). In the United States, land use contributed more to virus discovery than in other regions—urbanized land, urbanization of cropland, and growth of urbanized land alone had a relative contribution of 47.9%. It is possible that land use change in the US is driving both the emergence of novel viruses and their discovery, as has been

suggested for Heartland virus (*Mansfield et al., 2017*; *Savage et al., 2013*) and several hantaviruses (*Hassell et al., 2017*).

Climate had less influence on human-infective RNA virus discovery in all three regions in comparison to other predictors, in contrast to virus discovery at a global scale (*Zhang et al., 2020*). The underlying reason may be that the proportion of vector-borne viruses—whose distribution and abundance is strongly associated with the impact of climate on vector populations (*Li et al., 2014*)—in all three regions (United States: 23.2%; China: 21.3%; Africa: 27.1%) were less than that in the world (41.7%) (*Figure 3*). Vector-borne viruses tend to have more restricted global ranges, so are less likely to appear in a study of any one region (*Zhang et al., 2020*).

In addition, a relatively smaller proportion of strictly zoonotic viruses in three regions (United States: 30.5%; China: 16.3%; Africa: 33.6%) than that in the world (58.7%) (*Figure 2*) made biodiversity contribute less to virus discovery in the three regions than in the world (*Zhang et al., 2020*). With exposure to a higher density of mammals played a slightly larger role in virus discovery in Africa than in China and the United States (*Appendix 3—figure 9* to *Appendix 3—figure 11*).

Our discovery probability maps for 2010–2019 in three regions captured most historical hotspots, though several small new areas in central-eastern and southwestern United States, eastern and western China, as well as northern Africa would also make greater contribution to virus discovery (*Figure 6*). Our model has a good predictive ability, given 84% (37/44) new virus species in 2010–2019 were discovered in high-risk areas we have defined—85% percentiles of discovery probability within each region. Further, 35% (13/37) of those viruses discovered in high-risk areas since 2010 were discovered at the potential new hotspots where there had not been any virus discoveries in the past.

Our subgroup analyses distinguishing viruses firstly discovered in regions and those that had been discovered elsewhere in the world suggested in both the United States and Africa, discoveries of viruses firstly discovered in regions were more likely to be associated with climatic and biodiversity variables while discoveries of viruses had been discovered elsewhere in the world were more likely to be associated with socio-economic variables. This is plausible, again because after a novel virus was discovered elsewhere in the world, it is usually areas with a higher socio-economic level that first capture the virus in the local region.

This study had limitations. First, one common problem for data collected from literature review is the time lag between virus discovery and publication, in which case the virus data are likely to be matched to covariates in later decades. Second, we acknowledge that it is possible we have not identified the earliest report for some well-known viruses such as yellow fever virus, measles virus, especially in the post-vaccination era. Third, we were unable to identify robust and comprehensive data for all three regions on virus discovery effort (e.g. government transparency, laboratory infrastructure and technology), although we interpret GDP and university count as being an indirect measure of resources available for this activity. Previous studies have tried to use the bibliographic data to correct for the discovery effort (; ). However, this strategy worked less well for our data as the frequency of published paper from virus-related scientific journals has only a weak link to publications on novel human-infective RNA virus (*Appendix 3—figure 1*).

The study adds to our previous study (*Zhang et al., 2020*) in several ways. First, we firstly construct data sets of human-infective RNA virus discovery reflecting the viral richness in three broad regions of the world. Second, we reduced the heterogeneity of the predictors by focusing on regions, including those predictors reflecting the research effort. Research effort is less variable within restricted regions and therefore has less effect on virus detection. This implies our predicted hotspots stand closer to the virus geographic distribution in nature. Third, the predicted hotspots derived from regional analysis have a higher precision than at a global scale, for example, specific areas in the United States and China were identified as hotspots from regional analysis, rather than the whole eastern area from the global analysis. This helps target areas for future surveillance.

In conclusion, a heterogeneous pattern of virus discovery-driver relationships was identified across three regions and the globe. Within regions virus discovery is driven more by land use and socio-economic variables; climate and biodiversity variables are consistently less important predictors than at a global scale. We mapped with good accuracy that in 2010–2019 three regions where human-infective RNA viruses had previously been discovered would continue to be the discovery hotspots, but in addition, several new areas in each region would make great contribution to virus discovery. Results from the study could guide active surveillance for new human-infective viruses in high-risk areas.

## Acknowledgements

We thank Liam Brierley (University of Liverpool, UK) for validating the data sets. We would like to thank all reviewers and editors (Benn Sartorius, Ben Cooper, George Perry etc.) for their constructive comments and suggestions.

## Additional information

### Funding

| Funder | Grant reference number | Author |
|---|---|---|
| Darwin Trust of Edinburgh | | Feifei Zhang |
| European Union's Horizon 2020 research and innovation programme | 874735 | Mark EJ Woolhouse |

The funders had no role in study design, data collection and interpretation, or the decision to submit the work for publication.

### Author contributions

Feifei Zhang, Conceptualization, Data curation, Formal analysis, Investigation, Methodology, Resources, Software, Validation, Visualization, Writing - original draft, Writing – review and editing; Margo Chase-Topping, Methodology, Supervision, Writing – review and editing; Chuan-Guo Guo, Data curation, Methodology, Software, Validation, Writing – review and editing; Mark EJ Woolhouse, Conceptualization, Funding acquisition, Methodology, Project administration, Supervision, Writing - original draft, Writing – review and editing

### Author ORCIDs

Feifei Zhang http://orcid.org/0000-0002-3718-243X

### Decision letter and Author response

Decision letter https://doi.org/10.7554/eLife.72123.sa1
Author response https://doi.org/10.7554/eLife.72123.sa2

## Additional files

### Supplementary files

• Transparent reporting form

### Data availability

The authors confirm that all data or the data sources are provided in the paper and its Supplementary Materials. The final datasets and codes used for the analyses are available via figshare at https://doi.org/10.6084/m9.figshare.15101979.

The following dataset was generated:

| Author(s) | Year | Dataset title | Dataset URL | Database and Identifier |
|---|---|---|---|---|
| Zhang F | 2021 | Supporting data and R scripts for: Predictors of human RNA virus discovery in the United States, China and Africa | https://doi.org/10.6084/m9.figshare.15101979 | figshare, 10.6084/m9.figshare.15101979 |

The following previously published dataset was used:

| Author(s) | Year | Dataset title | Dataset URL | Database and Identifier |
|---|---|---|---|---|
| Woolhouse MEJ, Brierley L | 2017 | Epidemiological characteristics of human-infective RNA viruses | http://dx.doi.org/10.7488/ds/2265 | Edinburgh DataShare, 10.7488/ds/2265 |

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

# Appendix 1

**Appendix 1—table 1.** Summary of the human-infective RNA virus data sets in the United States, Africa, and China.

| Species | Original discovery year | United States Reported? | Discovery year | location | Lat | Lon | China Reported? | Discovery year | location | Lat | Lon | Africa Reported? | Discovery year | location | Lat | Lon |
|---|---|---|---|---|---|---|---|---|---|---|---|---|---|---|---|---|
| Argentinian mammarenavirus | 1958 | No | | | | | No | | | | | No | | | | |
| Brazilian mammarenavirus | 1994 | Yes *Barry et al., 1995* | 1995 | New Haven, Connecticut | 41.31 | --72.93 | No | | | | | No | | | | |
| Cali mammarenavirus | 1971 | Yes *Buchmeier et al., 1974* | 1974 | Houston, Texas | 29.76 | --95.37 | No | | | | | No | | | | |
| Chapare mammarenavirus | 2008 | No | | | | | No | | | | | No | | | | |
| Guanarito mammarenavirus | 1991 | No | | | | | No | | | | | No | | | | |
| Lassa mammarenavirus | 1970 | Yes *Buckley and Casals, 1970* | 1970 | New Haven, Connecticut | 41.31 | --72.93 | No | | | | | Yes *Buckley and Casals, 1970* | 1970 | Lassa, Borno State, Nigeria | 10.69 | 13.27 |
| Lujo mammarenavirus | 2009 | No | | | | | No | | | | | Yes *Briese et al., 2009* | 2009 | Lusaka, Zambia | --15.39 | 28.32 |
| Lymphocytic choriomeningitis mammarenavirus | 1934 | Yes *Armstrong and Lillie, 1934* | 1934 | St. Louis county, Missouri | 38.61 | --90.41 | No | | | | | No | | | | |
| Machupo mammarenavirus | 1964 | No | | | | | No | | | | | No | | | | |
| Mobala mammarenavirus | 1985 | No | | | | | No | | | | | Yes *Georges et al., 1985* | 1985 | Bouboui and Gomoka village, Boali town, Central African Republic | 4.89 | 18.14 |
| Whitewater Arroyo mammarenavirus | 2000 | Yes *Enserink, 2000* | 2000 | Alameda County, California | 37.60 | --121.72 | No | | | | | No | | | | |
| Mamastrovirus 1 | 1975 | Yes *Oshiro et al., 1981* | 1981 | Martin County, California | 40.22 | --123.10 | Yes *Xu et al., 1981* | 1981 | Guangzhou, Guangdong | 23.13 | 113.26 | Yes *Dowling and Wynne, 1981* | 1981 | Lebowa, South Africa | --23.5 | 29.5 |
| Mamastrovirus 6 | 2008 | Yes *Finkbeiner et al., 2009c* | 2009 | St. Louis, Missouri | 38.63 | --90.20 | Yes *Chu et al., 2010* | 2010 | Hong Kong | 22.40 | 114.11 | Yes *Kapoor et al., 2009* | 2009 | Maiduguri, Borno State, Nigeria | 11.83 | 13.15 |
| Mamastrovirus 8 | 2009 | Yes *Finkbeiner et al., 2009a* | 2009 | St. Louis, Missouri | 38.63 | --90.20 | Yes *Wang et al., 2013* | 2013 | Nanjing, Jiangsu and Lanzhou, Gansu | 31.95 | 118.78 | Yes *Kapoor et al., 2009* | 2009 | Maiduguri, Borno State, Nigeria | 11.83 | 13.15 |
| Mamastrovirus 9 | 2009 | Yes *Finkbeiner et al., 2009b* | 2009 | Accomack and Northampton Counties, Virginia | 37.71 | --75.81 | Yes *Tao et al., 2019* | 2019 | Jinan, Shandong | 36.68 | 117.11 | Yes *Kapoor et al., 2009* | 2009 | Maiduguri, Borno State, Nigeria | 11.83 | 13.15 |
| Mammalian 1 orthobornavirus | 1985 | Yes *Rott et al., 1985* | 1985 | Philadelphia, Pennsylvania | 39.95 | --75.17 | Yes *Chen et al., 1999* | 1999 | Taiwan | 23.70 | 120.96 | Yes *Bode et al., 1992* | 1992 | Rural area of East Africa | --1.28 | 34.53 |
| Mammalian 2 orthobornavirus | 2015 | No | | | | | No | | | | | No | | | | |
| Norwalk virus | 1972 | Yes *Kapikian et al., 1972* | 1972 | Norwalk, Ohio | 41.24 | --82.62 | Yes *Fang et al., 1995* | 1995 | Henan | 33.88 | 113.48 | Yes *Taylor et al., 1993* | 1993 | Pretoria, Gauteng province, South Africa | --25.75 | 28.23 |
| Sapporo virus | 1980 | Yes *Nakata et al., 1988* | 1988 | Houston, Texas | 29.76 | --95.37 | Yes *Nakata et al., 1988* | 1988 | Shanghai | 31.23 | 121.47 | Yes *Wolfaardt et al., 1997* | 1997 | Pretoria, Gauteng province, South Africa | --25.75 | 28.23 |
| Vesicular exanthema of swine virus | 1998 | Yes *Smith et al., 1998* | 1998 | Corvallis, Oregon | 44.56 | --123.26 | No | | | | | No | | | | |
| Alphacoronavirus 1 | 2007 | No | | | | | No | | | | | No | | | | |
| Human coronavirus 229E | 1966 | Yes *Hamre and Procknow, 1966* | 1966 | Chicago, Illinois | 41.88 | --87.63 | Yes *Virus Research Group of Kun Number 323 Unit, The Chinese People's Liberation Army, 1975* | 1975 | Kunming, Yunnan | 25.07 | 102.68 | Yes *Hays and Myint, 1998* | 1998 | Kumasi, Ghana | 6.70 | --1.62 |
| Human coronavirus NL63 | 2004 | Yes *Esper et al., 2005* | 2005 | New Haven, Connecticut | 41.31 | --72.93 | Yes *Chan et al., 2005* | 2005 | Hong Kong | 22.40 | 114.11 | Yes *Smuts et al., 2008* | 2008 | Cape Town, Western Cape Province, South Africa | --33.90 | 18.57 |
| Betacoronavirus 1 | 1967 | Yes *McIntosh et al., 1967* | 1967 | Bethesda, Maryland | 38.98 | --77.09 | Yes *Chan et al., 2005* | 2005 | Hong Kong | 22.40 | 114.11 | Yes *Venter et al., 2011* | 2011 | Pretoria, Gauteng province, South Africa | --25.75 | 28.23 |
| Human coronavirus HKU1 | 2005 | Yes *Esper et al., 2006* | 2006 | New Haven, Connecticut | 41.31 | --72.92 | Yes *Woo et al., 2005* | 2005 | Hong Kong | 22.40 | 114.11 | Yes *Venter et al., 2011* | 2011 | Pretoria, Gauteng province, South Africa | --25.75 | 28.23 |

*Appendix 1—table 1 Continued on next page*

*Appendix 1—table 1 Continued*

| Species | Original discovery year | United States | | | | | China | | | | | Africa | | | | |
|---|---|---|---|---|---|---|---|---|---|---|---|---|---|---|---|---|
| | | Reported? | Discovery year | location | Lat | Lon | Reported? | Discovery year | location | Lat | Lon | Reported? | Discovery year | location | Lat | Lon |
| Middle East respiratory syndrome-related coronavirus | 2012 | Yes* *Bialek et al., 2014* | 2014 | Lake county, Indiana | 41.45 | --87.37 | Yes* *Gao and Song, 2015* | 2015 | Huizhou, Guangdong | 23.09 | 114.40 | Yes* *Abroug et al., 2014* | 2014 | Monastir, Tunisia | 35.79 | 10.82 |
| Severe acute respiratory syndrome-related coronavirus | 2003 | Yes* *Charles M, 2003* | 2003 | Atlanta, Georgia | 33.75 | --84.39 | Yes *Peiris et al., 2003a* | 2003 | Hong Kong | 22.40 | 114.11 | Yes *Chiu et al., 2004* | 2004 | Pretoria, Gauteng province, South Africa | --25.75 | 28.23 |
| Human torovirus (been abolished) | 1984 | No | | | | | No | | | | | No | | | | |
| Bundibugyo ebolavirus | 2008 | No | | | | | No | | | | | Yes *Smuts et al., 2008* | 2008 | Bundibugyo and Kikyo town, Bundibugyo District, Western Uganda | 0.71 | 30.06 |
| Reston ebolavirus | 1991 | Yes *Miranda et al., 1991* | 1991 | Reston, Fairfax County, Virginia | 38.96 | --77.35 | No | | | | | No | | | | |
| Sudan ebolavirus | 1977 | No | | | | | No | | | | | Yes *Bowen et al., 1977* | 1977 | Maridi, South Sudan | 4.91 | 29.45 |
| Tai Forest ebolavirus | 1995 | No | | | | | No | | | | | Yes *Le Guenno et al., 1995* | 1995 | Abidjan, Cote-d'Ivoire | 5.36 | --4.01 |
| Zaire ebolavirus | 1977 | No | | | | | No | | | | | Yes *Johnson et al., 1977* | 1977 | Yambuku village, Democratic Republic of the Congo | 2.83 | 22.22 |
| Marburg marburgvirus | 1968 | Yes* *Centers for Disease Control and Prevention, 2009* | 2009 | Denver county, Colorado | 39.55 | --105.78 | No | | | | | Yes *Gear et al., 1975* | 1975 | Johannesburg, South Africa | --26.20 | 27.90 |
| Aroa virus | 1971 | No | | | | | No | | | | | No | | | | |
| Bagaza virus | 2009 | No | | | | | No | | | | | No | | | | |
| Banzi virus | 1959 | No | | | | | No | | | | | Yes *Smithburn et al., 1959* | 1959 | Maponde's Kraal(Usutu river), South Africa | --26.52 | 31.67 |
| Cacipacore virus | 2011 | No | | | | | No | | | | | No | | | | |
| Dengue virus | 1907 | Yes *Lavinder and Francis, 1914* | 1914 | Savannah, Georgia | 32.02 | --81.12 | Yes *Clarke et al., 1967* | 1967 | Southwest Taiwan | 23.06 | 120.59 | Yes *Edington, 1927* | 1927 | Durban, KwaZulu-Natal Province, South Africa | --29.86 | 31.02 |
| Edge Hill virus | 1985 | No | | | | | No | | | | | No | | | | |
| Gadgets Gully virus | 1991 | No | | | | | No | | | | | No | | | | |
| Ilheus virus | 1947 | No | | | | | No | | | | | No | | | | |
| Japanese encephalitis virus | 1933 | Yes* *Perex-Pina and Merikangas, 1953* | 1953 | Waltham, Massachusetts | 42.38 | --71.24 | Yes *Yen, 1941* | 1941 | Beijing | 40.01 | 116.41 | Yes *Simon-Loriere et al., 2017* | 2017 | Cunene, Angola | --16.28 | 15.28 |
| Kokobera virus | 1964 | No | | | | | No | | | | | No | | | | |
| Kyasanur forest disease virus | 1957 | No | | | | | Yes *Wang et al., 2009* | 2009 | Hengduanshan Mountain, Yunnan | 27.50 | 99.00 | Yes *Andayi et al., 2014* | 2014 | Djibouti, Republic of Djibouti | 11.57 | 43.15 |
| Langat virus | 1956 | No | | | | | No | | | | | No | | | | |
| Louping ill virus | 1934 | Yes *Rivers and Schwentker, 1934* | 1934 | New York | 40.71 | --74.01 | No | | | | | No | | | | |
| Murray Valley encephalitis virus | 1952 | No | | | | | No | | | | | No | | | | |
| Ntaya virus | 1952 | No | | | | | No | | | | | Yes *Smithburn, 1952* | 1952 | Bwamba county, Uganda | 0.75 | 30.02 |
| Omsk hemorrhagic fever virus | 1948 | No | | | | | No | | | | | No | | | | |
| Powassan virus | 1959 | Yes *Goldfield et al., 1973* | 1973 | Middlesex County, New Jersey | 40.54 | --74.37 | No | | | | | No | | | | |
| Rio Bravo virus | 1962 | Yes *Suklin et al., 1962* | 1962 | Dallas city, Texas | 32.78 | --96.80 | No | | | | | No | | | | |
| Saint Louis encephalitis virus | 1933 | Yes *Webster and Fite, 2009* | 1933 | St. Louis City, Missouri | 38.63 | --90.20 | No | | | | | No | | | | |
| Tembusu virus | 1975 | No | | | | | Yes *Tang et al., 2013* | 2013 | Shandong | 36.40 | 118.77 | No | | | | |

*Appendix 1—table 1 Continued on next page*

*Appendix 1—table 1 Continued*

| Species | Original discovery year | United States | | | | | China | | | | | Africa | | | | |
|---|---|---|---|---|---|---|---|---|---|---|---|---|---|---|---|---|
| | | Reported? | Discovery year | location | Lat | Lon | Reported? | Discovery year | location | Lat | Lon | Reported? | Discovery year | location | Lat | Lon |
| Tick-borne encephalitis virus | 1938 | Yes* *Cruse et al., 1979* | 1979 | Cleveland, Ohio | 41.51 | --81.69 | Yes *Wang and Zhao, 1956* | 1956 | Bali village, Wuchang, Heilongjiang | 44.91 | 127.16 | No | | | | |
| Uganda S virus | 1952 | No | | | | | No | | | | | Yes *Dick and Haddow, 1952* | 1952 | Bwamba county, Uganda | 0.75 | 30.02 |
| Usutu virus | 2009 | No | | | | | No | | | | | No | | | | |
| Wesselsbron virus | 1957 | No | | | | | No | | | | | Yes *Smithburn et al., 1957* | 1957 | Lake Simbu region, Maputaland, KwaZulu-Natal, South Africa | --27.36 | 32.32 |
| West Nile virus | 1940 | Yes *Nash et al., 2001* | 2001 | New York | 40.71 | --74.01 | Yes *Li et al., 2013* | 2013 | Jiashi County, Xinjiang | 39.58 | 77.18 | Yes *Smithburn et al., 1940* | 1940 | Omogo, West Nile district, Uganda | 0.42 | 33.21 |
| Yellow fever virus | 1901 | Yes *Guiteras, 1904* | 1904 | Laredo, Texas | 27.51 | --99.51 | Yes* *Chen and Lu, 2016* | 2016 | Beijing | 40.01 | 116.41 | Yes *Stokes et al., 1928* | 1928 | Larteh, Ghana | 5.94 | --0.07 |
| Zika virus | 1952 | Yes* *Foy et al., 2011* | 2011 | Northern Colorado | 39.55 | --105.78 | Yes* *Sun et al., 2016* | 2016 | Gan County, Ganzhou city, Jiangxi | 25.86 | 115.02 | Yes *Dick, 1952* | 1952 | Zika, Uganda | 0.12 | 32.53 |
| Hepacivirus C | 1989 | Yes *Choo et al., 1989* | 1989 | Emeryville, California | 37.83 | --122.29 | Yes *Xu et al., 1990a* | 1990 | Qidong county, Jiangsu | 31.88 | 121.72 | Yes *Kew et al., 1990* | 1990 | Johannesburg, South Africa | --26.20 | 27.90 |
| Pegivirus C | 1995 | Yes *Simons et al., 1995* | 1995 | Chapel Hill, North Carolina; Rochester, Minnesota; Dallas, Texas | 35.91 | --79.06 | Yes *Wang et al., 1996* | 1996 | Beijing | 40.01 | 116.41 | Yes *Simons et al., 1995* | 1995 | Cairo, Egypt | 30.04 | 31.24 |
| Pegivirus H | 2015 | Yes *Kapoor et al., 2015* | 2015 | New York city, New York | 40.71 | --74.01 | Yes *Wang et al., 2018* | 2018 | Guangzhou, Guangdong | 23.13 | 113.26 | Yes *Rodgers et al., 2019* | 2019 | Ebolowa, Cameroon | 2.92 | 11.15 |
| Pestivirus A | 1988 | Yes *Yolken et al., 1989* | 1989 | Whiteriver, Arizona | 33.83 | --109.97 | No | | | | | Yes *Giangaspero et al., 1988* | 1988 | Zambia | --13.13 | 27.85 |
| Andes orthohantavirus | 1996 | No | | | | | No | | | | | No | | | | |
| Bayou orthohantavirus | 1995 | Yes *Morzunov et al., 1995* | 1995 | Louisiana | 30.98 | --91.96 | No | | | | | No | | | | |
| Black creek canal orthohantavirus | 1995 | Yes *Ravkov et al., 1995* | 1995 | Miami-Dade County, Florida | 25.76 | --80.34 | No | | | | | No | | | | |
| Choclo orthohantavirus | 2000 | No | | | | | No | | | | | No | | | | |
| Dobrava-Belgrade orthohantavirus | 1992 | No | | | | | No | | | | | No | | | | |
| Hantaan orthohantavirus | 1978 | No | | | | | Yes *Lee et al., 1980* | 1980 | Zhejiang | 29.14 | 119.79 | No | | | | |
| Laguna Negra orthohantavirus | 1997 | No | | | | | No | | | | | No | | | | |
| Puumala orthohantavirus | 1980 | No | | | | | No | | | | | No | | | | |
| Sangassou orthohantavirus | 2010 | No | | | | | No | | | | | Yes *Klempa et al., 2010* | 2010 | Sangassou village, Macenta district, Forest Guinea | 8.24 | --9.32 |
| Seoul orthohantavirus | 1982 | Yes *Forthal et al., 1987* | 1987 | Mississippi | 32.57 | --89.88 | Yes *Song et al., 1982* | 1982 | Jiangsu | 33.14 | 119.79 | Yes *Tomori et al., 1986* | 1986 | Jos, Nigeria | 9.90 | 8.86 |
| Sin Nombre orthohantavirus | 1993 | Yes *Nichol et al., 1993* | 1993 | New Mexico | 34.52 | --105.87 | No | | | | | No | | | | |
| Thailand orthohantavirus | 2006 | No | | | | | No | | | | | No | | | | |
| Thottopalayam thottimvirus | 2007 | No | | | | | No | | | | | No | | | | |
| Tula orthohantavirus | 1996 | No | | | | | No | | | | | No | | | | |
| Orthohepevirus A | 1983 | Yes* *De Cock et al., 1987* | 1987 | Los Angeles County, California | 34.05 | --118.24 | Yes *Huang et al., 1989* | 1989 | Kashi county, Kashi city, Xinjiang | 39.46 | 75.99 | Yes *Belabbes et al., 1985* | 1985 | Medea town, Algeria | 36.26 | 2.75 |
| Orthohepevirus C | 2018 | No | | | | | Yes *Sridhar et al., 2018* | 2018 | Hong Kong | 22.40 | 114.11 | No | | | | |
| Crimean-Congo haemorrhagic fever orthonairovirus | 1967 | No | | | | | Yes *Yen et al., 1985* | 1985 | Bachu, southern Xinjiang | 39.79 | 78.55 | Yes *Simpson et al., 1967* | 1967 | Kisangani, Tshopo province, Democratic Republic of the Congo | 0.53 | 25.19 |

*Appendix 1—table 1 Continued on next page*

*Appendix 1—table 1 Continued*

| Species | Original discovery year | United States | | | | | China | | | | | Africa | | | | |
|---|---|---|---|---|---|---|---|---|---|---|---|---|---|---|---|---|
| | | Reported? | Discovery year | location | Lat | Lon | Reported? | Discovery year | location | Lat | Lon | Reported? | Discovery year | location | Lat | Lon |
| Dugbe orthonairovirus | 1969 | No | | | | | No | | | | | Yes *Causey et al., 1969* | 1969 | Ibadan, Nigeria | 7.35 | 3.88 |
| Nairobi sheep disease orthonairovirus | 1969 | No | | | | | No | | | | | Yes *Morrill et al., 1991* | 1991 | Mombasa; Malindi; and Kilifi, Coast Province, Kenya | --3.34 | 39.57 |
| Thiafora orthonairovirus | 1989 | No | | | | | No | | | | | No | | | | |
| Influenza A virus | 1933 | Yes *Francis and Magill, 1935* | 1935 | Philadelphi, Pennsylvania | 39.95 | --75.17 | Yes *Chang and Chiang, 1950* | 1950 | Beijing | 40.01 | 116.41 | Yes *Isaacs and Andrews, 1951* | 1951 | Johannesburg, South Africa and Cape Town, South Africa | --26.20 | 27.90 |
| Influenza B virus | 1940 | Yes *Francis, 1940* | 1940 | Irvington village, Greenburgh town, Westchester County, New York | 41.03 | --73.87 | Yes *Wen and Chu, 1957* | 1957 | Beijing | 40.01 | 116.41 | Yes *Montefiore et al., 1970* | 1970 | Arusha, Arusha Region, Tanzania | --3.37 | 36.69 |
| Influenza C virus | 1950 | Yes *Francis et al., 1950* | 1950 | Ann Arbor city, Michigan | 42.28 | --83.74 | Yes *Zhang, 1957* | 1957 | Beijing | 40.01 | 116.41 | Yes *Joosting et al., 1968* | 1968 | Johannesburg, South Africa | --26.20 | 27.90 |
| Dhori thogotovirus | 1985 | No | | | | | No | | | | | No | | | | |
| Thogoto thogotovirus | 1969 | No | | | | | No | | | | | Yes *Causey et al., 1969* | 1969 | Ibadan, Nigeria | 7.35 | 3.88 |
| Avian orthoavulavirus 1 | 1943 | Yes *Burnet, 1943* | 1943 | Washington, D. C. | 38.91 | --77.04 | No | | | | | No | | | | |
| Hendra henipavirus | 1995 | No | | | | | No | | | | | No | | | | |
| Nipah henipavirus | 1999 | No | | | | | No | | | | | No | | | | |
| Canine morbillivirus | 1955 | Yes *Karzon, 1955* | 1955 | Buffalo, New York | 42.89 | --78.88 | No | | | | | No | | | | |
| Measles morbillivirus | 1911 | Yes *Goldberger and Anderson, 1911* | 1911 | Washington, D. C. | 38.91 | --77.04 | Yes *Tang et al., 1958* | 1958 | Beijing | 40.01 | 116.41 | Yes *Baylet et al., 1963* | 1963 | Dakar, Senegal | 14.72 | 17.47 |
| Human respirovirus 1 | 1958 | Yes *Chanock et al., 1958* | 1958 | Washington, D. C. | 38.91 | --77.04 | Yes *Chen et al., 1964* | 1964 | Zhejiang | 29.14 | 119.79 | Yes *Taylor-Robinson and Tyrrell, 1963* | 1963 | Cape Town, Western Cape Province, South Africa | --33.90 | 18.57 |
| Human respirovirus 3 | 1958 | Yes *Chanock et al., 1958* | 1958 | Washington, D. C. | 38.91 | --77.04 | Yes *Yu et al., 1987* | 1987 | Guangzhou, Guangdong | 23.13 | 113.26 | Yes *Taylor-Robinson and Tyrrell, 1963* | 1963 | Cape Town, Western Cape Province, South Africa | --33.90 | 18.57 |
| Achimota pararubulavirus 2 | 2013 | No | | | | | No | | | | | Yes *Baker et al., 2013* | 2013 | Volta, Ghana | 6.05 | 0.37 |
| Human orthorubulavirus 2 | 1956 | Yes *Chanock, 1956* | 1956 | Cincinnati, Ohio | 39.10 | --84.51 | Yes *Pathogen biology research group, Jiangsu new medical college, 1975* | 1975 | Nanjing, Jiangsu | 31.95 | 118.78 | Yes *Balestrieri et al., 1967* | 1967 | Accra, Ghana | 5.60 | --0.19 |
| Human orthorubulavirus 4 | 1960 | Yes *Johnson et al., 1960* | 1960 | Bethesda, Maryland | 38.98 | --77.09 | Yes *Lau et al., 2005* | 2005 | Hong Kong | 22.40 | 114.11 | Yes *Niang et al., 2010* | 2010 | Ndiop village, Sine Saloum region, Senegal | 15.18 | 16.74 |
| Mammalian orthorubulavirus 5 | 1959 | Yes *Schultz and Habel, 1959* | 1959 | Stanford, California | 37.42 | --122.17 | No | | | | | No | | | | |
| Menangle pararubulavirus | 1998 | No | | | | | No | | | | | No | | | | |
| Mumps orthorubulavirus | 1934 | Yes *Johnson and Goodpasture, 1934* | 1934 | Nashville, Tennessee | 36.16 | --86.78 | Yes *Wang et al., 1958* | 1958 | Beijing | 40.01 | 116.41 | Yes *Bayer and Gear, 1955* | 1955 | Johannesburg, South Africa | --26.20 | 27.90 |
| Simian orthorubulavirus | 1968 | No | | | | | No | | | | | No | | | | |
| Sosuga pararubulavirus | 2014 | No | | | | | No | | | | | Yes *Albariño et al., 2014* | 2014 | - | 3.76 | 32.82 |
| Tioman pararubulavirus | 2007 | No | | | | | No | | | | | No | | | | |
| Bunyamwera orthobunyavirus | 1946 | Yes *Work, 1964* | 1964 | Southern Florida | 26.92 | --81.21 | No | | | | | Yes *Smithburn et al., 1946* | 1946 | Bwamba County, Uganda | 0.75 | 30.02 |
| Bwamba orthobunyavirus | 1941 | No | | | | | No | | | | | Yes *Smithburn et al., 1941* | 1941 | Bwamba county, Western Province of Uganda | 0.75 | 30.02 |
| California encephalitis orthobunyavirus | 1952 | Yes *Hammon and Reeves, 1952* | 1952 | Kern county, California | 35.49 | --118.86 | Yes *Gu et al., 1984* | 1984 | Longhua, Shanghai | 31.22 | 121.43 | Yes *Bardos and Sefcovicova, 1961* | 1961 | Uganda | 1.37 | 32.29 |
| Caraparu orthobunyavirus | 1961 | No | | | | | No | | | | | No | | | | |

*Appendix 1—table 1 Continued on next page*

*Appendix 1—table 1 Continued*

| Species | Original discovery year | United States | | | | | China | | | | | Africa | | | | |
|---|---|---|---|---|---|---|---|---|---|---|---|---|---|---|---|---|
| | | Reported? | Discovery year | location | Lat | Lon | Reported? | Discovery year | location | Lat | Lon | Reported? | Discovery year | location | Lat | Lon |
| Catu orthobunyavirus | 1961 | No | | | | | No | | | | | No | | | | |
| Guama orthobunyavirus | 1961 | No | | | | | No | | | | | No | | | | |
| Guaroa orthobunyavirus | 1959 | No | | | | | No | | | | | No | | | | |
| Kairi orthobunyavirus | 1967 | No | | | | | No | | | | | No | | | | |
| Madrid orthobunyavirus | 1964 | No | | | | | No | | | | | No | | | | |
| Marituba orthobunyavirus | 1961 | No | | | | | No | | | | | No | | | | |
| Nyando orthobunyavirus | 1965 | No | | | | | No | | | | | Yes *Williams et al., 1965* | 1965 | Kisumu, Kenya | --0.09 | 34.77 |
| Oriboca orthobunyavirus | 1961 | No | | | | | No | | | | | No | | | | |
| Oropouche orthobunyavirus | 1961 | No | | | | | No | | | | | No | | | | |
| Patois orthobunyavirus | 1972 | No | | | | | No | | | | | No | | | | |
| Shuni orthobunyavirus | 1975 | No | | | | | No | | | | | Yes *Moore et al., 1975* | 1975 | Ibadan, Nigeria | 7.38 | 3.95 |
| Tacaiuma orthobunyavirus | 1967 | No | | | | | No | | | | | No | | | | |
| Wyeomyia orthobunyavirus | 1965 | No | | | | | No | | | | | No | | | | |
| Candiru phlebovirus | 1983 | No | | | | | No | | | | | No | | | | |
| Punta Toro phlebovirus | 1970 | No | | | | | No | | | | | No | | | | |
| Rift Valley fever phlebovirus | 1931 | No | | | | | Yes* *Liu et al., 2016* | 2016 | Beijing | 40.01 | 116.41 | Yes *Daubney et al., 1931* | 1931 | Rift Valley of Kenya Colony | --0.28 | 36.07 |
| Sandfly fever Naples phlebovirus | 1944 | No | | | | | No | | | | | Yes *Sabin, 1951* | 1951 | Cairo, Egypt | 30.04 | 31.24 |
| Heartland banyangvirus | 2012 | Yes *McMullan et al., 2012* | 2012 | Andrew and Nodaway Counties, Missouri | 39.82 | --94.59 | No | | | | | No | | | | |
| Huaiyangshan banyangvirus | 2011 | No | | | | | Yes *Zhang et al., 2011* | 2011 | Huaiyangshan | 31.37 | 115.39 | No | | | | |
| Uukuniemi phlebovirus | 1970 | No | | | | | No | | | | | No | | | | |
| Human picobirnavirus | 1988 | Yes *Grohmann et al., 1993* | 1993 | Atlanta, Georgia | 33.75 | --84.39 | Yes *Rosen et al., 2000* | 2000 | Lulong County, Hebei | 39.94 | 116.94 | No | | | | |
| Equine rhinitis A virus | 1962 | No | | | | | No | | | | | No | | | | |
| Foot-and-mouth disease virus | 1965 | No | | | | | Yes *Luo et al., 1999* | 1999 | Guangzhou | 23.13 | 113.26 | Yes *Donia and Youssef, 2002* | 2002 | Alexandria Governorate, Egypt | 30.74 | 29.74 |
| Cardiovirus A | 1947 | Yes *Jonkers, 1961* | 1961 | New Orleans, Louisiana | 29.95 | --90.07 | Yes *Feng et al., 2015* | 2015 | Changchun, Jilin | 43.87 | 125.34 | Yes *Dick and Best, 1948* | 1948 | Entebbe, Uganda | 0.05 | 32.46 |
| Cardiovirus B | 1963 | Yes *Jones et al., 2007* | 2007 | San Diego, California | 32.72 | 117.16 -- | Yes *Cheng et al., 2009a* | 2009 | Lanzhou, Gansu | 36.06 | 103.79 | Yes *Zoll et al., 2009* | 2009 | Cameroon | 5.03 | 12.40 |
| Cosavirus A | 2008 | No | | | | | Yes *Dai et al., 2010* | 2010 | Shanghai | 31.23 | 121.47 | Yes *Kapusinszky et al., 2012* | 2012 | Maiduguri, Borno State, Nigeria | 11.83 | 13.15 |
| Cosavirus B | 2008 | No | | | | | Yes *Yang et al., 2016* | 2016 | Zhenjiang, Jiangsu | 32.19 | 119.43 | No | | | | |
| Cosavirus D | 2008 | No | | | | | No | | | | | Yes *Kapusinszky et al., 2012* | 2012 | Maiduguri, Borno State, Nigeria | 11.83 | 13.15 |
| Cosavirus E | 2008 | No | | | | | No | | | | | Yes *Kapusinszky et al., 2012* | 2012 | Maiduguri, Borno State, Nigeria | 11.83 | 13.15 |
| Cosavirus F | 2012 | No | | | | | No | | | | | No | | | | |
| Enterovirus A | 1949 | Yes *Sickles and Dalldorf, 1949* | 1949 | New York | 43.30 | --74.22 | Yes *Xiao et al., 1985* | 1985 | Tianjin | 39.34 | 117.36 | Yes *Bayer and Gear, 1955* | 1955 | Johannesburg, South Africa | --26.20 | 27.90 |

*Appendix 1—table 1 Continued on next page*

*Appendix 1—table 1 Continued*

| Species | Original discovery year | United States | | | | | China | | | | | Africa | | | | |
|---|---|---|---|---|---|---|---|---|---|---|---|---|---|---|---|---|
| | | Reported? | Discovery year | location | Lat | Lon | Reported? | Discovery year | location | Lat | Lon | Reported? | Discovery year | location | Lat | Lon |
| Enterovirus B | 1949 | Yes *Sickles and Dalldorf, 1949* | 1949 | Wilmington | 39.74 | --75.54 | Yes *Wu et al., 1960* | 1960 | Fuzhou, Fujian | 26.07 | 119.30 | Yes *Patz et al., 1953* | 1953 | Middelburg, Transvaal, South Africa | --25.77 | 29.46 |
| Enterovirus C | 1909 | Yes *Flexner and Lewis, 1909* | 1909 | New York city, New York | 40.71 | --74.01 | Yes *Yen and Hsü, 1941* | 1941 | Bejing | 39.90 | 116.41 | Yes *Hudson and Lennette, 1933* | 1933 | Monrovia, Liberia | 6.29 | 10.76 |
| Enterovirus D | 1967 | Yes *Schieble et al., 1967* | 1967 | Berkeley, California | 37.87 | 122.27 | Yes *Shanghai Eye and Skin Disease Prevention and Treatment Institute, 1979* | 1979 | Shanghai | 31.23 | 121.47 | Yes *Mirkovic et al., 1973* | 1973 | Morocco | 31.79 | --7.09 |
| Enterovirus E | 1961 | Yes *Moscovivci et al., 1961* | 1961 | Denver, Colorado | 39.74 | 104.99 | No | | | | | No | | | | |
| Enterovirus H | 1965 | No | | | | | No | | | | | No | | | | |
| Rhinovirus A | 1953 | Yes *Price, 1956* | 1956 | Baltimore, Maryland | 39.29 | --76.61 | Yes *Guangzhou Institute of Medicine and Health, 1975* | 1975 | Guangzhou, Guangdong | 23.13 | 113.26 | Yes *Taylor-Robinson, 1963* | 1963 | Cape Town, Western Cape Province, South Africa | --33.90 | 18.57 |
| Rhinovirus B | 1960 | Yes *Hamre and Procknow, 1961* | 1961 | Chicago, Illinois | 41.88 | --87.63 | Yes *Xiang et al., 2008* | 2008 | Beijing | 40.01 | 116.41 | Yes *Briese et al., 2008* | 2008 | Pretoria, Gauteng province, South Africa | --25.75 | 28.23 |
| Rhinovirus C | 2006 | Yes *Lamson et al., 2006* | 2006 | New York city, New York | 40.71 | --74.01 | Yes *Lau et al., 2007* | 2007 | Hong Kong | 22.40 | 114.11 | Yes *Briese et al., 2008* | 2008 | Pretoria, Gauteng province, South Africa | --25.75 | 28.23 |
| Erbovirus A | 2005 | No | | | | | No | | | | | No | | | | |
| Hepatovirus A | 1973 | Yes *Feinstone et al., 1973* | 1973 | Bethesda, Maryland | 38.98 | --77.09 | Yes *Microbiology Research Group of Shanghai First Medical College and Laboratory of Shanghai Sixth People's Hospital, 1978* | 1978 | Shanghai | 31.23 | 121.47 | Yes *Szmuness et al., 1977* | 1977 | Dakar, Senegal | 14.72 | 17.47 |
| Aichivirus A | 1991 | Yes *Chhabra et al., 2013* | 2013 | Cincinnati, Ohio | 39.10 | --84.51 | Yes *Yang et al., 2009* | 2009 | Shanghai | 31.23 | 121.47 | Yes *Sdiri-Loulizi et al., 2008* | 2008 | Monastir, Tunisia | 35.77 | 10.82 |
| Parechovirus A | 1958 | Yes *Ramoz-alverz and Sabin, 1958* | 1958 | Cincinnati, Ohio | 39.10 | --84.51 | Yes *Shan et al., 2009* | 2009 | Shanghai | 31.23 | 121.47 | Yes *Kapusinszky et al., 2012* | 2012 | Ouagadougou, Burkina Faso | 12.24 | --1.56 |
| Parechovirus B | 2003 | No | | | | | No | | | | | No | | | | |
| Salivirus A | 2009 | Yes *Greninger et al., 2009* | 2009 | Northern California | 38.84 | 120.90 | Yes *Shan et al., 2010* | 2010 | Shanghai | 31.23 | 121.47 | Yes *Li et al., 2009* | 2009 | Maiduguri, Borno State, Nigeria | 11.83 | 13.15 |
| Avian metapneumovirus | 2011 | Yes *Kayali et al., 2011* | 2011 | Memphis, Tennessee | 35.15 | --90.05 | No | | | | | No | | | | |
| Human metapneumovirus | 2001 | Yes *Falsey et al., 2003* | 2003 | Rochester, New York | 43.16 | --77.61 | Yes *Peiris et al., 2003b* | 2003 | Hong Kong | 22.40 | 114.11 | Yes *Madhi et al., 2003* | 2003 | Johannesburg, South Africa | --26.20 | 27.90 |
| Human orthopneumovirus | 1957 | Yes *Chanock et al., 1957* | 1957 | Baltimore, Maryland | 39.29 | --76.61 | Yes *Kun Number 323 Unit, the Chinese People's Liberation Army, 1975* | 1975 | Kunming, Yunnan | 25.07 | 102.68 | Yes *Doggett, 1965* | 1965 | Cape Town, Western Cape Province, South Africa | --33.90 | 18.57 |
| Colorado tick fever virus | 1946 | Yes *Florio et al., 1946* | 1946 | Denver, Colorado | 39.74 | 104.99 | Yes *Yang et al., 1996* | 1996 | Nanjing, Jiangsu | 31.95 | 118.78 | No | | | | |
| Eyach virus | 1980 | No | | | | | No | | | | | No | | | | |
| Corriparta virus | 1967 | No | | | | | No | | | | | No | | | | |
| Great Island virus | 1963 | No | | | | | No | | | | | No | | | | |
| Lebombo virus | 1975 | No | | | | | No | | | | | Yes *Moore et al., 1975* | 1975 | Ibadan, Nigeria | 7.38 | 3.95 |
| Orungo virus | 1976 | No | | | | | No | | | | | Yes *Tomori et al., 1976* | 1976 | Ibadan, Nigeria | 7.38 | 3.95 |
| Mammalian orthoreovirus | 1954 | Yes *Ramos-Alvarez and Sabin, 1954* | 1954 | Cincinnati, Ohio | 39.10 | --84.51 | Yes *Zhao et al., 1995* | 1995 | Xuzhou, Jiangsu | 34.26 | 117.19 | Yes *Malherbe et al., 1963* | 1963 | Johannesburg, South Africa | --26.20 | 27.90 |
| Nelson Bay orthoreovirus | 2007 | No | | | | | Yes* *Cheng et al., 2009b* | 2009 | Hong Kong | 22.40 | 114.11 | No | | | | |
| Rotavirus A | 1973 | Yes *Kapikian et al., 1976* | 1976 | Washington, D. C. | 38.90 | --77.04 | Yes *PaPa et al., 1979* | 1979 | Beijing | 40.01 | 116.41 | Yes *Tomori et al., 1976* | 1976 | Johannesburg, South Africa | --26.20 | 27.90 |
| Rotavirus B | 1984 | Yes *Eiden et al., 1985* | 1985 | Baltimore, Maryland | 39.29 | --76.61 | Yes *Hung et al., 1984* | 1984 | Jinzhou, Liaoning | 41.10 | 121.13 | Yes *Nakata et al., 1987* | 1987 | Kenya | --0.02 | 37.91 |
| Rotavirus C | 1986 | Yes *Jiang et al., 1995* | 1995 | Providence, Rhode Island | 41.82 | --71.41 | Yes *Qiao et al., 1999* | 1999 | Beijing | 40.01 | 116.41 | Yes *Sebata and Steele, 1999* | 1999 | Pretoria, Gauteng province, South Africa | --25.75 | 28.23 |

*Appendix 1—table 1 Continued on next page*

*Appendix 1—table 1 Continued*

| Species | Original discovery year | United States | | | | | China | | | | | Africa | | | | |
|---|---|---|---|---|---|---|---|---|---|---|---|---|---|---|---|---|
| | | Reported? | Discovery year | location | Lat | Lon | Reported? | Discovery year | location | Lat | Lon | Reported? | Discovery year | location | Lat | Lon |
| Rotavirus H | 1987 | No | | | | | Yes *Wang et al., 1987* | 1987 | Huaihua, Hunan Province | 27.55 | 109.96 | No | | | | |
| Banna virus | 1990 | No | | | | | Yes *Xu et al., 1990b* | 1990 | Xishuangbanna, Yunnan Province | 21.90 | 100.80 | No | | | | |
| Primate T-lymphotropic virus 1 | 1980 | Yes *Poiesz et al., 1980* | 1980 | Bethesda, Maryland | 38.98 | --77.09 | Yes *Hung et al., 1984* | 1984 | Shenyang, Liaoing | 41.80 | 123.38 | Yes *Williams et al., 1984* | 1984 | Ibadan, Nigeria | 7.38 | 3.95 |
| Primate T-lymphotropic virus 2 | 1982 | Yes *Kalyanaraman et al., 1982* | 1982 | Seattle, Washington | 47.61 | --122.33 | Yes *Ma et al., 2013* | 2013 | Henan and Hubei | 32.21 | 112.96 | Yes *Delaporte et al., 1991* | 1991 | Franceville, Gabon | --1.63 | 13.60 |
| Primate T-lymphotropic virus 3 | 2005 | No | | | | | No | | | | | Yes *Calattini et al., 2005* | 2005 | Océan department, South Province, Cameroon | 2.50 | 10.50 |
| Human immunodeficiency virus 1 | 1983 | Yes *Safai et al., 1984* | 1984 | Washington, D. C. | 38.90 | --77.04 | Yes *Chang et al., 1986* | 1986 | Hong Kong | 22.40 | 114.11 | Yes *Brun-Vézinet et al., 1984* | 1984 | Kisangani, Tshopo province, Democratic Republic of the Congo | 0.53 | 25.19 |
| Human immunodeficiency virus 2 | 1986 | Yes* *Centers for Disease Control, 1988* | 1988 | New Jersey | 40.06 | --74.41 | Yes* *Yan et al., 2000* | 2000 | Fuzhou, Fujian | 26.07 | 119.30 | Yes *Kanki et al., 1986* | 1986 | Dakar, Senegal | 14.72 | --17.47 |
| Simian immunodeficiency virus | 1992 | Yes *Khabbaz et al., 1992* | 1992 | Atlanta, Georgia | 33.75 | --84.39 | No | | | | | Yes *Calattini et al., 2005* | 2005 | Cameroon | 7.37 | 12.35 |
| Central chimpanzee simian foamy virus | 2012 | No | | | | | No | | | | | Yes *Rua et al., 2012* | 2012 | Near Dja Nature Reserves, Southern Cameroon | 4.50 | 13.50 |
| Eastern chimpanzee simian foamy virus | 1971 | No | | | | | No | | | | | Yes *Achong et al., 1971* | 1971 | Kenya | --0.02 | 37.91 |
| Grivet simian foamy virus | 1997 | No | | | | | No | | | | | No | | | | |
| Guenon simian foamy virus | 2012 | No | | | | | No | | | | | Yes *Rua et al., 2012* | 2012 | Near Iolodrof, Southern Cameroon | 3.23 | 10.73 |
| Taiwanese macaque simian foamy virus | 2002 | No | | | | | Yes *Huang et al., 2012* | 2012 | Yunnan | 25.18 | 101.86 | No | | | | |
| Australian bat lyssavirus | 1998 | No | | | | | No | | | | | No | | | | |
| Duvenhage lyssavirus | 1971 | No | | | | | No | | | | | Yes *Meredith et al., 1971* | 1971 | Pretoria, Gauteng province, South Africa | --25.75 | 28.23 |
| European bat Yeslyssavirus | 1989 | No | | | | | No | | | | | No | | | | |
| European bat 2 lyssavirus | 1986 | No | | | | | No | | | | | No | | | | |
| Irkut lyssavirus | 2013 | No | | | | | Yes *Liu et al., 2013* | 2013 | Tonghua county, Jilin | 41.68 | 125.76 | No | | | | |
| Mokola lyssavirus | 1972 | No | | | | | No | | | | | Yes *Familusi et al., 1972* | 1972 | Ibadan, Nigeria | 7.38 | 3.95 |
| Rabies lyssavirus | 1903 | Yes *Black and Powers, 1910* | 1910 | Southern California | 34.57 | --116.76 | Yes *Wu, 1981* | 1981 | Beijing | 40.01 | 116.41 | Yes *Wilhelm and Alexis, 1933* | 1933 | Carolina, Mpumalanga, South Africa | --26.07 | 30.12 |
| Bas-Congo tibrovirus | 2012 | No | | | | | No | | | | | Yes *Grard et al., 2012* | 2012 | Mangala village, Boma Bungu Health Zone, Democratic Republic of Congo (DRC) | --4.04 | 21.76 |
| Ekpoma Yestibrovirus | 2015 | No | | | | | No | | | | | Yes *Stremlau et al., 2015* | 2015 | Irrua, Edo State, Nigeria | 6.74 | 6.22 |
| Ekpoma 2 tibrovirus | 2015 | No | | | | | No | | | | | Yes *Stremlau et al., 2015* | 2015 | Irrua, Edo State, Nigeria | 6.74 | 6.22 |
| Alagoas vesiculovirus | 1967 | No | | | | | No | | | | | No | | | | |
| Chandipura vesiculovirus | 1967 | No | | | | | No | | | | | No | | | | |
| Cocal vesiculovirus | 1964 | No | | | | | No | | | | | No | | | | |

*Appendix 1—table 1 Continued on next page*

*Appendix 1—table 1 Continued*

| Species | Original discovery year | United States | | | | | China | | | | | Africa | | | | |
|---|---|---|---|---|---|---|---|---|---|---|---|---|---|---|---|---|
| | | Reported? | Discovery year | location | Lat | Lon | Reported? | Discovery year | location | Lat | Lon | Reported? | Discovery year | location | Lat | Lon |
| Indiana vesiculovirus | 1958 | Yes *Patterson et al., 1958* | 1958 | Beltsville, Prince George's County, Maryland | 39.05 | --76.90 | No | | | | | No | | | | |
| Isfahan vesiculovirus | 1977 | No | | | | | No | | | | | No | | | | |
| Maraba vesiculovirus | 1984 | No | | | | | No | | | | | No | | | | |
| New Jersey vesiculovirus | 1950 | Yes *Hanson et al., 1950* | 1950 | Madison, Wisconsin | 43.07 | --89.40 | No | | | | | No | | | | |
| Piry vesiculovirus | 1974 | No | | | | | No | | | | | No | | | | |
| Barmah Forest virus | 1986 | No | | | | | No | | | | | No | | | | |
| Chikungunya virus | 1956 | Yes* *Centers for Disease Control and Prevention, 2006* | 2006 | Minnesota | 46.44 | --93.36 | Yes *Clarke et al., 1967* | 1967 | Southwest Taiwan | 23.06 | 120.59 | Yes *Ross, 1956* | 1956 | Newala district, Tanzania | --10.64 | 39.24 |
| Eastern equine encephalitis virus | 1938 | Yes *Howitt, 1938* | 1938 | Southwestern Massachusetts | 42.19 | --73.09 | No | | | | | No | | | | |
| Everglades virus | 1970 | Yes *Ehrenkranz et al., 1970* | 1970 | Homestead, Florida | 25.47 | --80.48 | No | | | | | No | | | | |
| Getah virus | 1966 | No | | | | | Yes *Li et al., 1992* | 1992 | Baoting County, Hainan | 18.98 | 109.83 | No | | | | |
| Highlands J virus | 2000 | Yes *Meehan et al., 2000* | 2000 | Florida | 27.66 | --81.52 | No | | | | | No | | | | |
| Madariaga virus | 1972 | No | | | | | No | | | | | No | | | | |
| Mayaro virus | 1957 | Yes* *Tesh et al., 1999* | 1999 | Ohio | 40.42 | --82.91 | No | | | | | No | | | | |
| Mosso das Pedras virus | 2013 | No | | | | | No | | | | | No | | | | |
| Mucambo virus | 1965 | No | | | | | No | | | | | No | | | | |
| Ndumu virus | 1961 | No | | | | | No | | | | | Yes *Kokernot et al., 1961* | 1961 | Ndumu, Maputaland, KwaZulu-Natal, South Africa | --26.93 | 32.26 |
| Onyong-nyong virus | 1961 | No | | | | | No | | | | | Yes *Williams and Woodall, 1961* | 1961 | Entebbe, Uganda | 0.05 | 32.46 |
| Pixuna virus | 1991 | No | | | | | No | | | | | No | | | | |
| Rio Negro virus | 1993 | No | | | | | No | | | | | No | | | | |
| Ross River virus | 1972 | No | | | | | Yes *Xu et al., 1999* | 1999 | Hainan | 19.16 | 109.94 | No | | | | |
| Semliki Forest virus | 1979 | No | | | | | No | | | | | Yes *Mathiot et al., 1990* | 1990 | Bangui, Central Africa | 4.36 | 18.58 |
| Sindbis virus | 1955 | No | | | | | No | | | | | Yes *Taylor et al., 1955* | 1955 | Cairo, Egypt | 30.04 | 31.24 |
| Tonate virus | 1976 | No | | | | | No | | | | | No | | | | |
| Una virus | 1963 | No | | | | | No | | | | | No | | | | |
| Venezuelan equine encephalitis virus | 1943 | Yes *Casals et al., 1943* | 1943 | New York | 40.71 | --74.01 | No | | | | | No | | | | |
| Western equine encephalitis virus | 1938 | Yes *Howitt, 1938* | 1938 | Fresno, California | 36.75 | --119.77 | No | | | | | No | | | | |
| Whataroa virus | 1964 | No | | | | | No | | | | | No | | | | |
| Rubella virus | 1942 | Yes *Habel, 1942* | 1942 | Washington, D. C. | 38.91 | --77.04 | Yes *He et al., 1979* | 1979 | Hangzhou, Zhejiang | 29.87 | 119.33 | Yes *Selzer, 1963* | 1963 | Cape Town, Western Cape Province, South Africa | --33.90 | 18.57 |
| Hepatitis delta virus | 1977 | Yes *Rizzetto et al., 1979* | 1979 | New Jersey | 40.06 | --74.41 | Yes *Rizzetto et al., 1980* | 1980 | Taipei, Taiwan | 24.96 | 121.51 | Yes *Crocchiolo et al., 1984* | 1984 | Harare, Zimbabwe | --17.83 | 31.03 |

Notes: Yes denotes the virus was ever discovered from the region; * denotes the virus was ever discovered from the region, but imported from other regions; No denotes the virus species has never been discovered from the region; The lat and long denote the coordidate of discovery points or centroids of polygons

**Appendix 1—table 2.** Resolution and covered grid cells for virus discovery data.

| | | Polygon data | | | Point data | Total |
|---|---|---|---|---|---|---|
| | | Country level | State/Province level | City/County level | | |
| United States | Virus counts | NA | 14 (14.7%) | 11 (11.6%) | 70 (73.7%) | 95 |
| | Gridded cell counts | NA | 189 | 12 | 72* | 273 |
| China | Virus counts | NA | 22 (27.5%) | 47 (58.7%) | 11 (13.8%) | 80 |
| | Gridded cell counts | NA | 161 | 70 | 12* | 243 |
| Africa | Virus counts | 7 (6.5%) | 5 (4.7%) | 15 (14.0%) | 80 (74.8%) | 107 |
| | Gridded cell counts | 307 | 22 | 17 | 80 | 426 |

*Grid cell counts here include viruses first detected in multiple points from the literature, NA, not applicable

**Appendix 1—table 3.** Model parameters.

| Model | Tree complexity | Learning rate | Bag fraction | No. of trees |
|---|---|---|---|---|
| United States | 2 | 0.0020 | 0.5 | 1430 |
| China | 2 | 0.0035 | 0.5 | 1473 |
| Africa | 2 | 0.0030 | 0.5 | 1446 |

**Appendix 1—table 4.** Model validation statistics for analyses in three regions.

| Model | % of deviance explained (95% quantiles) | ICC (95% quantiles) |
|---|---|---|
| United States | 50.5% (44.3%–56.8%) | 0.66 (0.60–0.70) |
| China | 42.0% (32.4%–50.8%) | 0.52 (0.41–0.60) |
| Africa | 42.4% (34.2%–50.0%) | 0.51 (0.44–0.62) |

ICC, intraclass correlation coefficient

## Appendix 2

We considered using bibliographic data to adjust for discovery effort, but rejected this strategy after some exploratory tests. *Jones et al., 2008* estimated the discovery effort for emerging infectious diseases (EID) by calculating the number of papers published by each country (denoted by the address for every author) in the Journal of Infectious Diseases (JID) since 1973. The hypothesis is that countries publishing more papers in JID are likely to discover more EID events. We tested whether this method worked for our analysis by plotting the relationship between published human-infective RNA virus count and total number of papers from all journals which published on human-infective RNA viruses in Web of Science (as of 21 Feb 2018). Both the total number of papers (*Appendix 3—figure 1A*) and total number of papers on viruses (*Appendix 3—figure 1B*) were weakly linked to the published human virus count in our database, though the number of papers did have a positive relationship with the number of papers on viruses (*Appendix 3—figure 1C*). We also noted that papers in JID (highlighted in blue in *Appendix 3—figure 1*) may not be able to fully explain the discovery efforts for newly discovered viruses. *Olival et al., 2017* adjust for the discovery effort by searching the number of publications for each of 586 virus species they have studied using a keyword search by virus name in PubMed and Web of Science. We found the results using this method were similar to that of *Jones et al., 2008*. *Allen et al., 2017* derived a different index for discovery bias, based on the spatial distribution of place names in peer-reviewed biomedical literature. The disadvantage of this method is that it may not represent the discovery effort, as many place names are not related to zoonotic viruses.

## Appendix 3

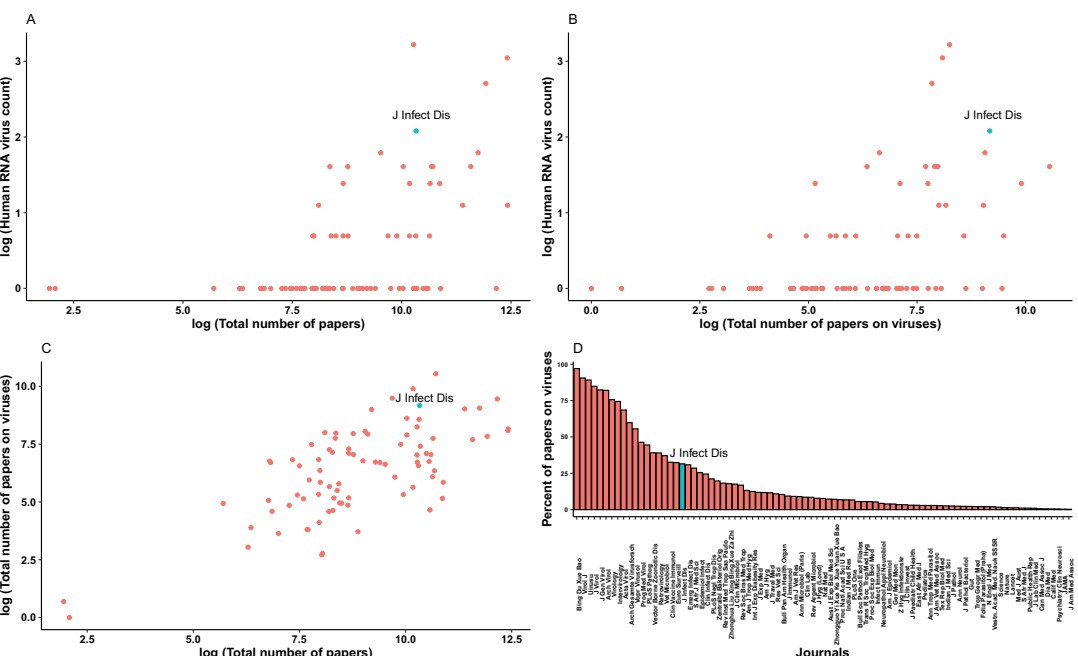

**Appendix 3—figure 1.** Relationship between published human-infective RNA virus count and total number of papers from the journals which published all human-infective RNA viruses in Web of Science. (A) Total number of papers vs. published human virus count; (B) Total number of papers on viruses vs. published human virus count; (C) Total number of papers vs. total number of papers on viruses; (D) Percent of papers on viruses in each journal. Journal of Infectious Diseases (JID) is highlighted in blue.

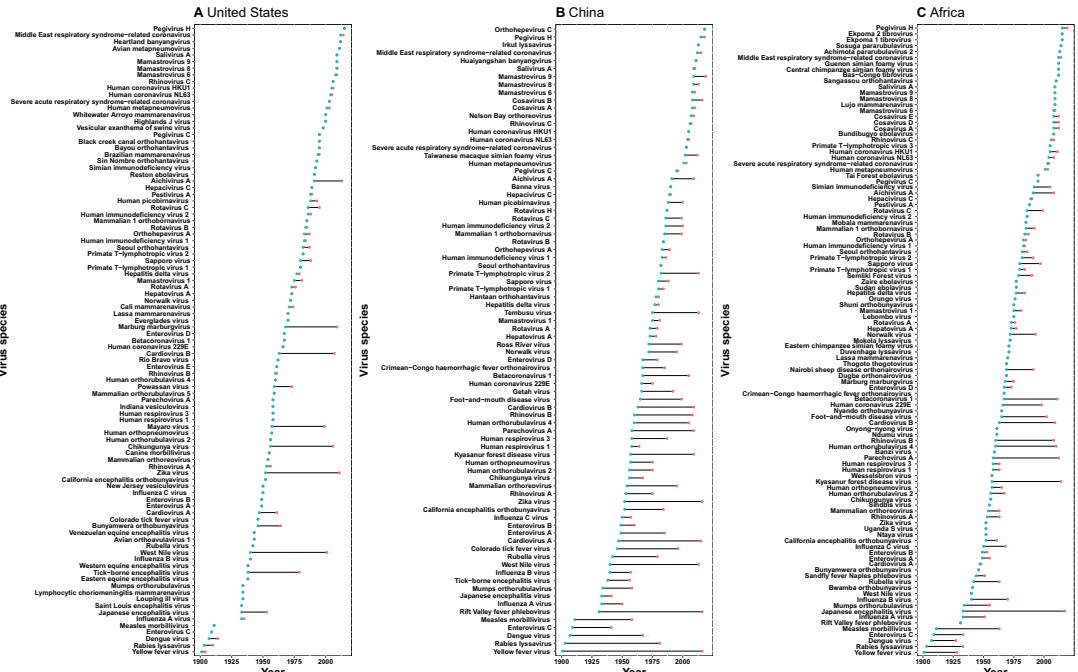

**Appendix 3—figure 2.** Time lag of human-infective RNA virus discovery between the three regions and the world. (A) United States. (B) China. (C) Africa. The blue dots represent the original discovery year of each virus in the world; the red dots represent the discovery year of each virus in three regions; and the segments between them represent the time lag.

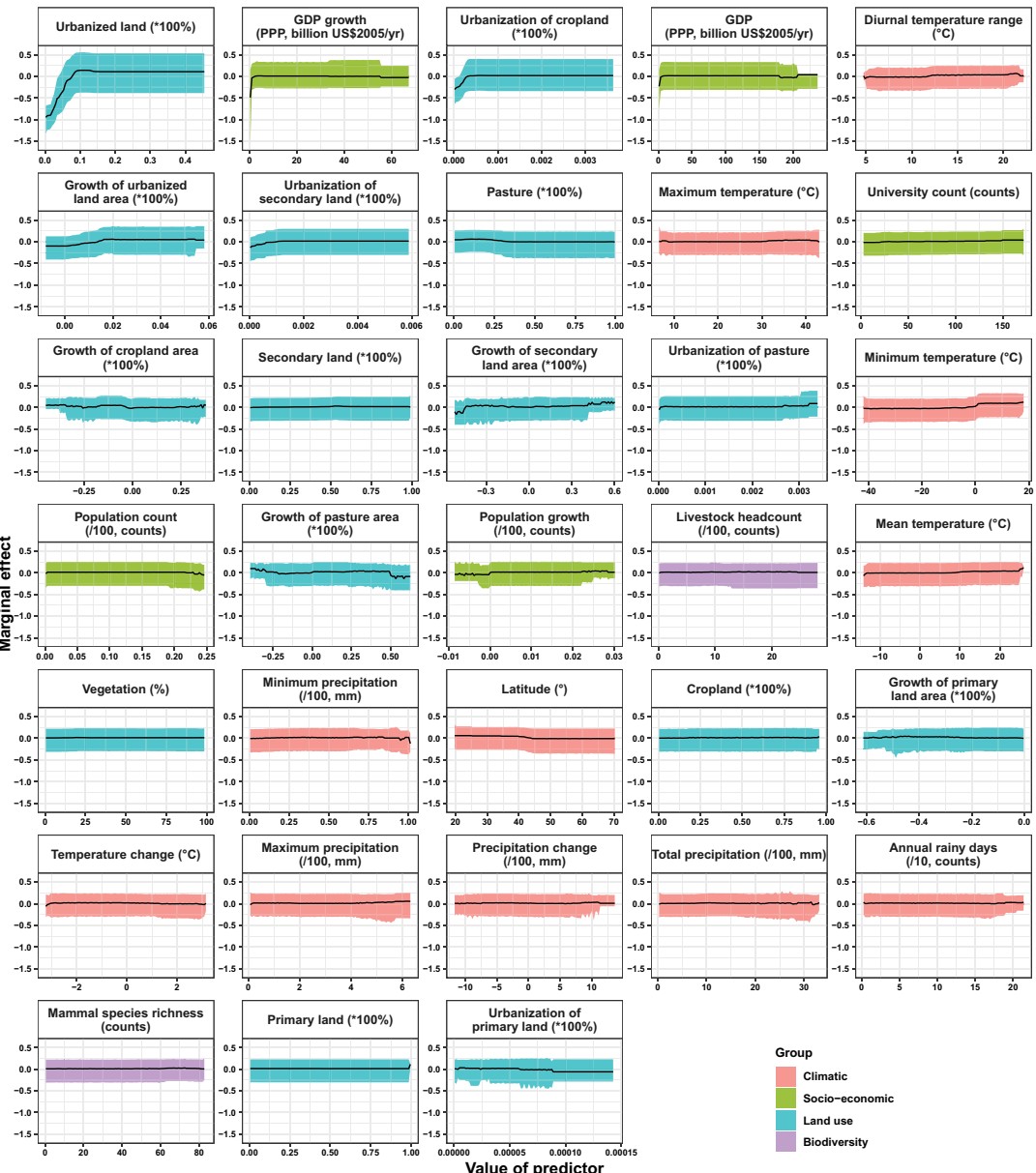

**Appendix 3—figure 3.** Partial dependence plots showing the influence on human-infective RNA virus discovery for all predictors in the Unites States. Partial dependence plots show the effect of an individual predictor over its range on the response after factoring out other predictors. Fitted lines represent the median (black) and 95% quantiles (coloured) based on 1000 replicated boosted regression tree models. Y axes are centred around the mean without scaling. X axes show the range of sampled values of predictors.

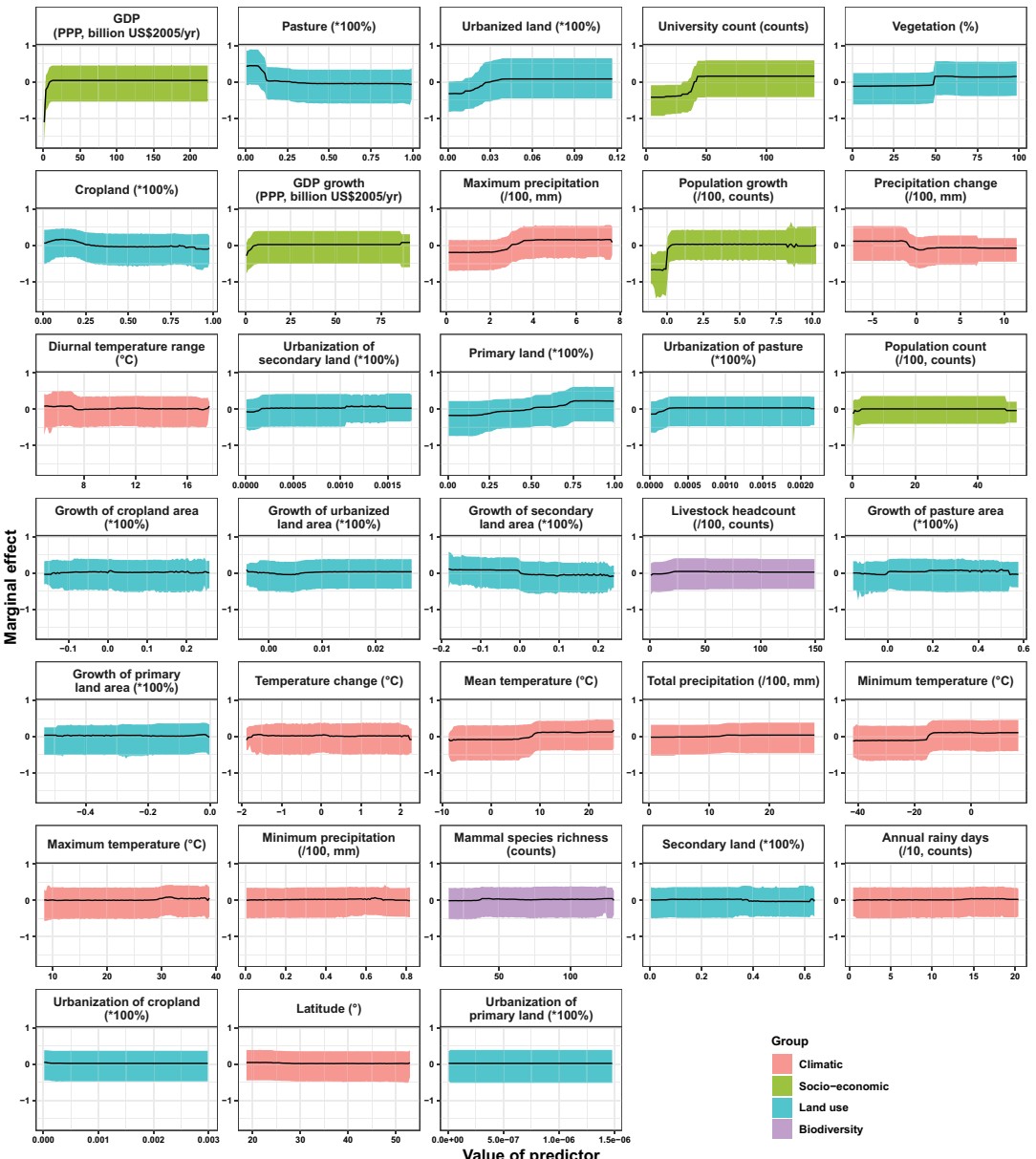

**Appendix 3—figure 4.** Partial dependence plots showing the influence on human-infective RNA virus discovery for predictors in China. Partial dependence plots show the effect of an individual predictor over its range on the response after factoring out other predictors. Fitted lines represent the median (black) and 95% quantiles (coloured) based on 1000 replicated boosted regression tree models. Y axes are centred around the mean without scaling. X axes show the range of sampled values of predictors.

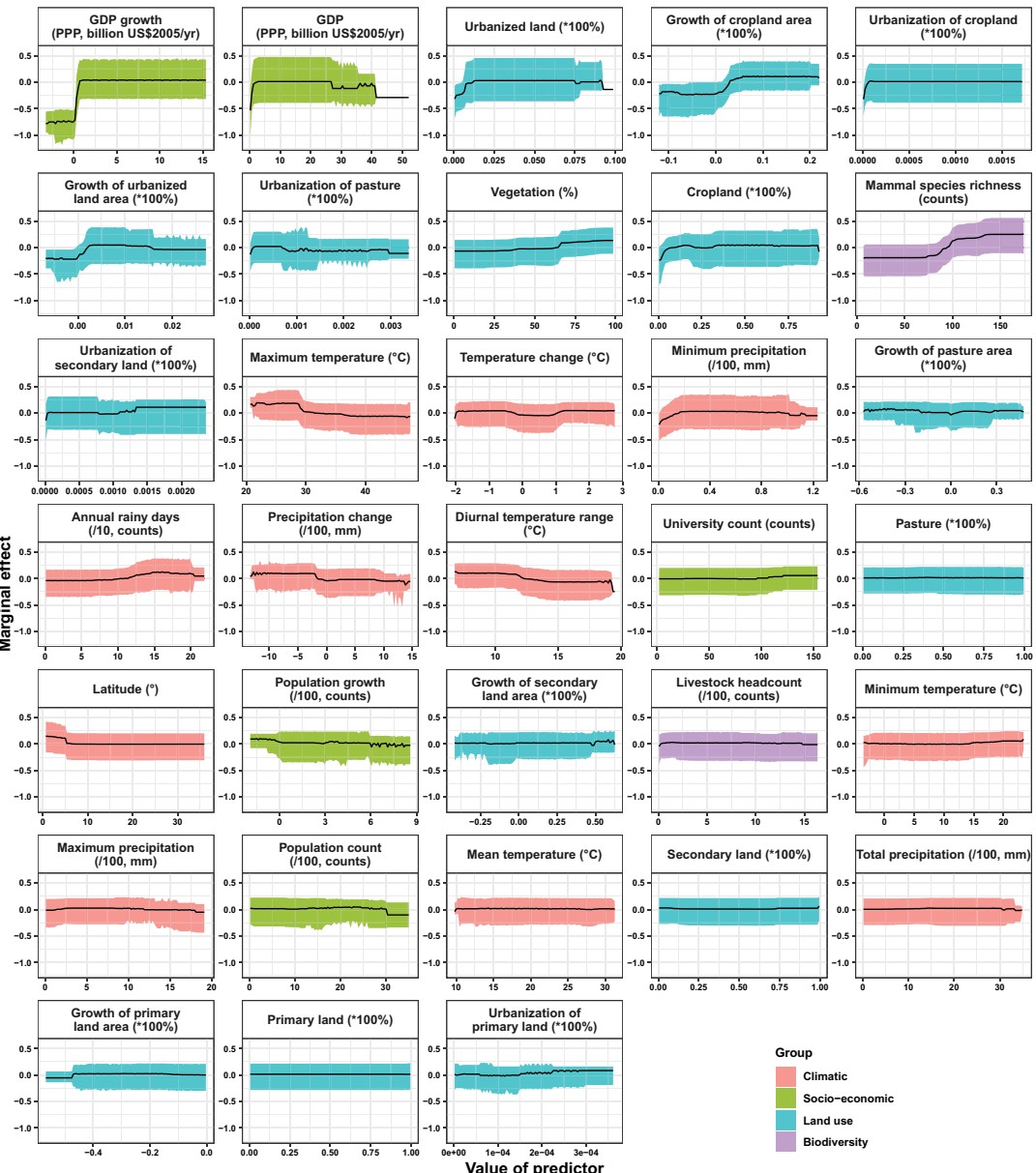

**Appendix 3—figure 5.** Partial dependence plots showing the influence on human-infective RNA virus discovery for all predictors in Africa. Partial dependence plots show the effect of an individual predictor over its range on the response after factoring out other predictors. Fitted lines represent the median (black) and 95% quantiles (coloured) based on 1000 replicated boosted regression tree models. Y axes are centred around the mean without scaling. X axes show the range of sampled values of predictors.

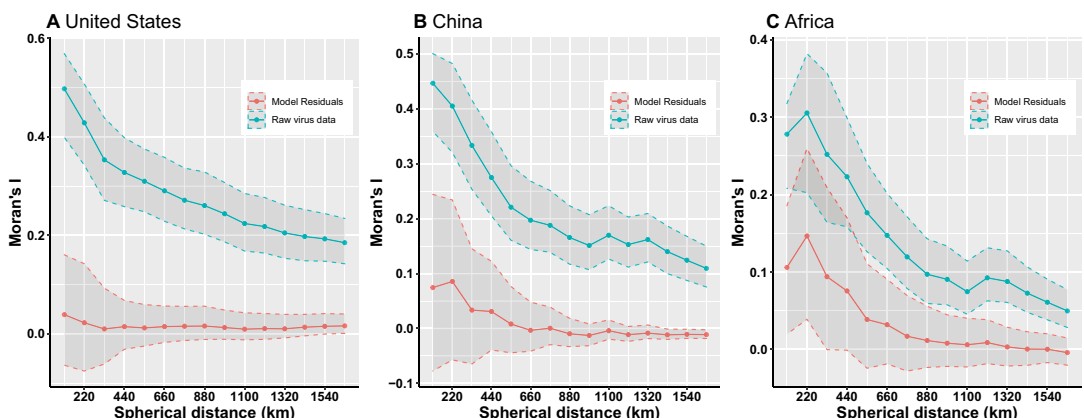

**Appendix 3—figure 6.** Moran's I across different spherical distances. (A) United States; (B) China; (C) Africa. The solid line and dots represented the median Moran's I value, and the grey area represented its 95% quantiles generated from 1000 samples (Blue: Raw virus data) or replicate boosted regression tree (BRT) models (Red: Model residuals). We used the fixed spherical distance as the neighbourhood weights—as there is no general consensus for selecting cut-off values, we chose spherical distances ranging from one time to fifteen times of distance of 1° grid cell at the equator, i.e. 110km to 1650km, considering the area of three regions. Our BRT models reduced Moran's I value from a range of 0.19–0.50 for the raw virus data to 0.009–0.04 for the model residuals in the United States (A), 0.11–0.45 to –0.01–0.09 in China (B), 0.05–0.31 to –0.004–0.15 in Africa (C), suggesting that BRT models with 33 predictors have adequately accounted for spatial autocorrelations in the raw virus data in all three regions.

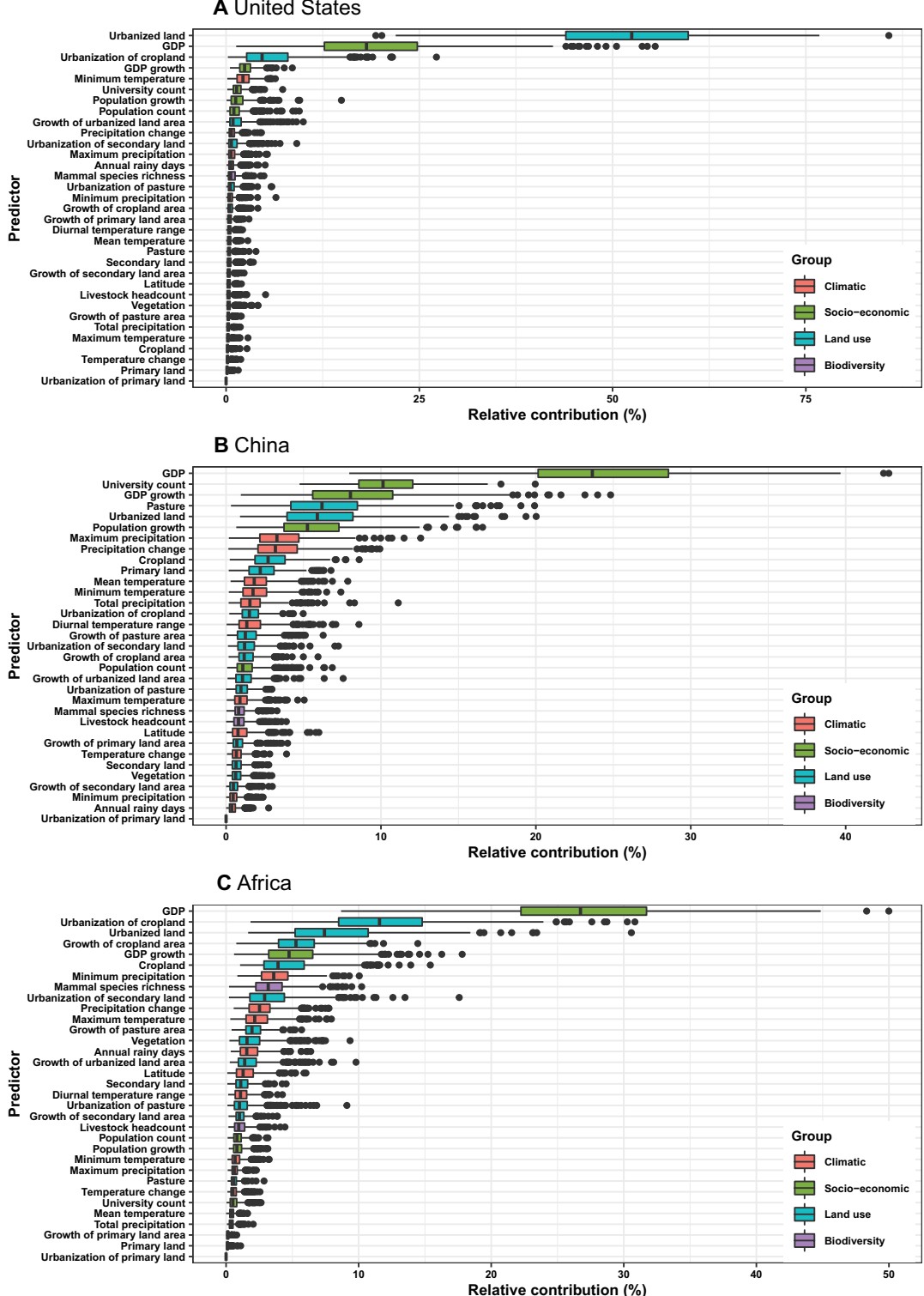

**Appendix 3—figure 7.** Relative contribution of predictors to human-infective RNA virus discovery in three regions. Virus discovery data were matched to time-varying covariate data by year. (A) United States. (B) China. (C) Africa. The boxplots show the median (black bar) and interquartile range (box) of the relative contribution across 1000 replicate boosted regression tree models, with whiskers indicating minimum and maximum and black dots indicating outliers.

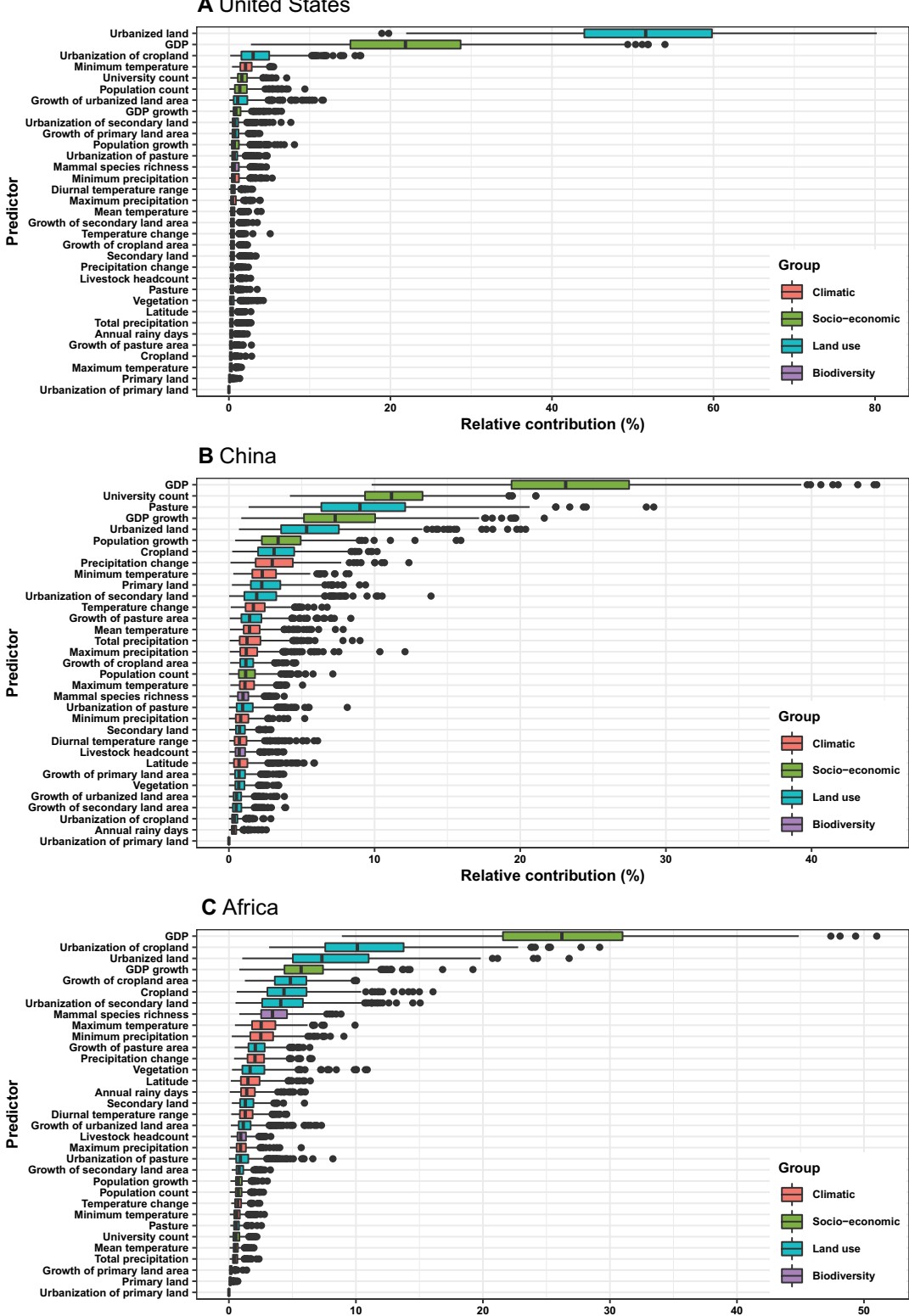

**Appendix 3—figure 8.** Relative contribution of predictors to human-infective RNA virus discovery in three regions. Virus discovery data at year t were matched to time-varying covariate data at year t-1. (A) United States. (B) China. (C) Africa. The boxplots show the median (black bar) and interquartile range (box) of the relative contribution across 1000 replicate boosted regression tree models, with whiskers indicating minimum and maximum and black dots indicating outliers.

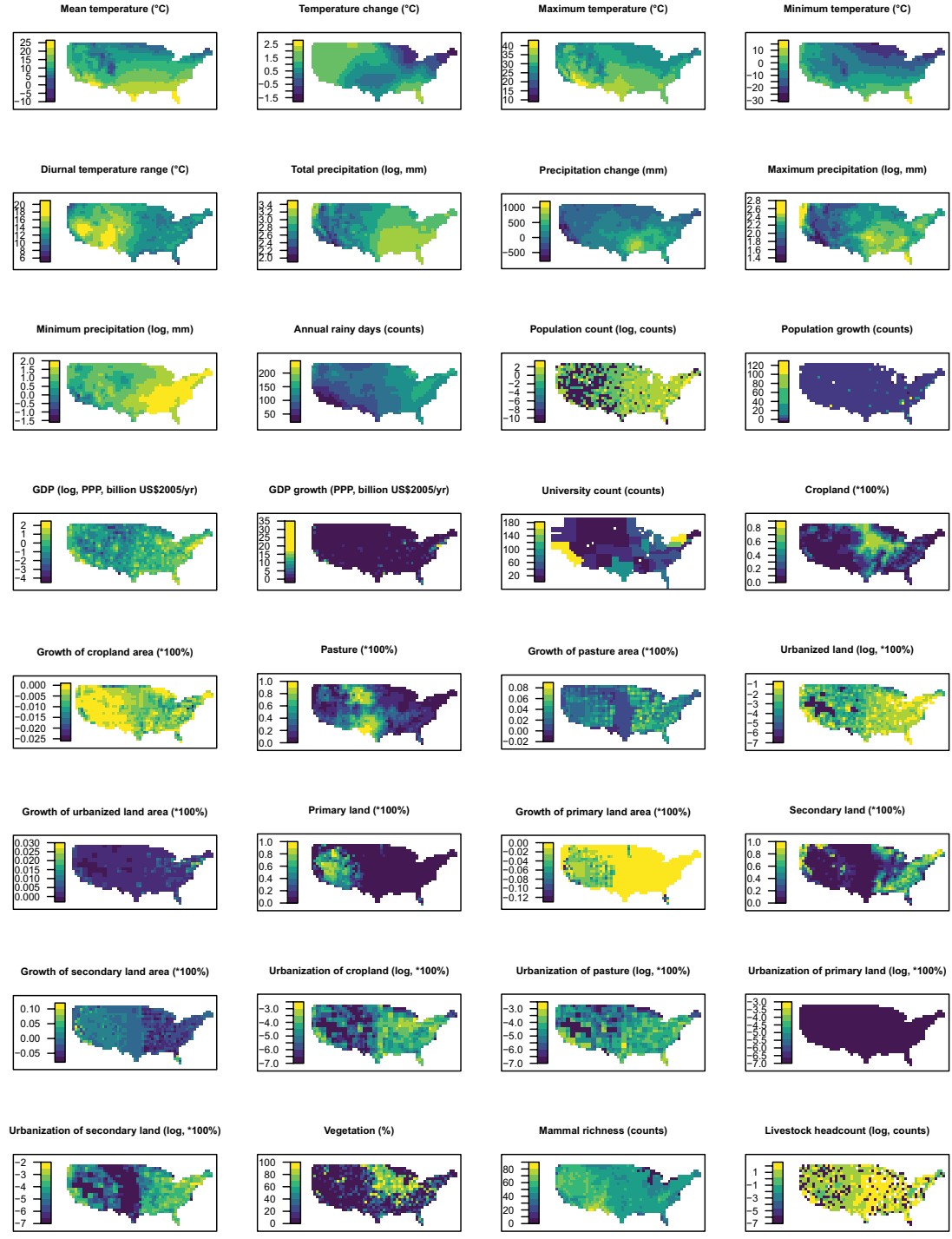

**Appendix 3—figure 9.** Distribution maps for 32 predictors in 2015 in the United States. The values of these explanatory variables and latitude in each grid cell were used to predict the virus discovery in the corresponding grid cell in the Unites States in 2010–2019. Explanatory variables were log transformed where necessary to get better visualization, not meaning they entered the model by logged values.

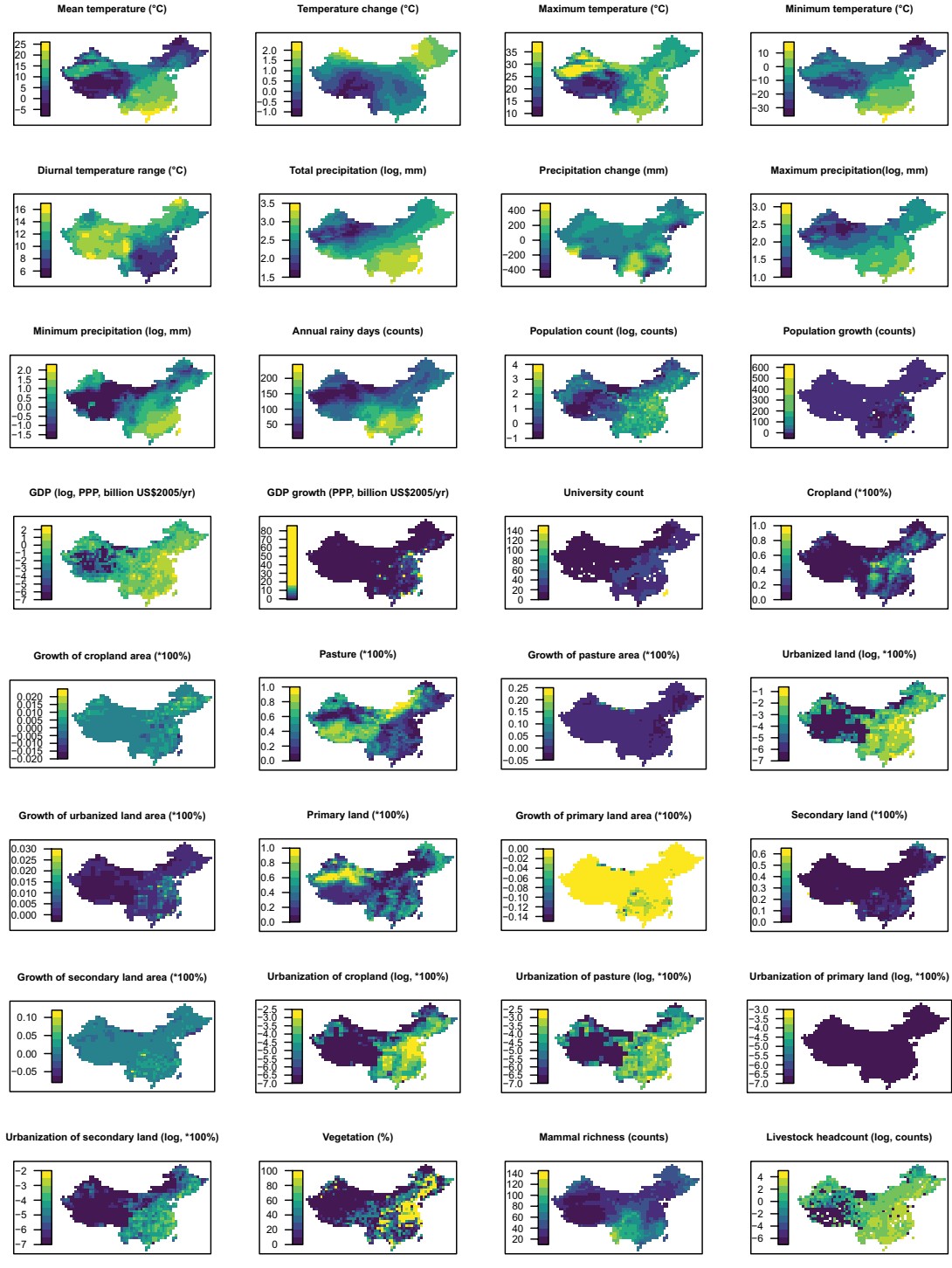

**Appendix 3—figure 10.** Distribution maps for 32 predictors in 2015 in China. The values of these explanatory variables and latitude in each grid cell were used to predict the virus discovery in the corresponding grid cell in China in 2010–2019. Explanatory variables were log transformed where necessary to get better visualization, not meaning they entered the model by logged values.

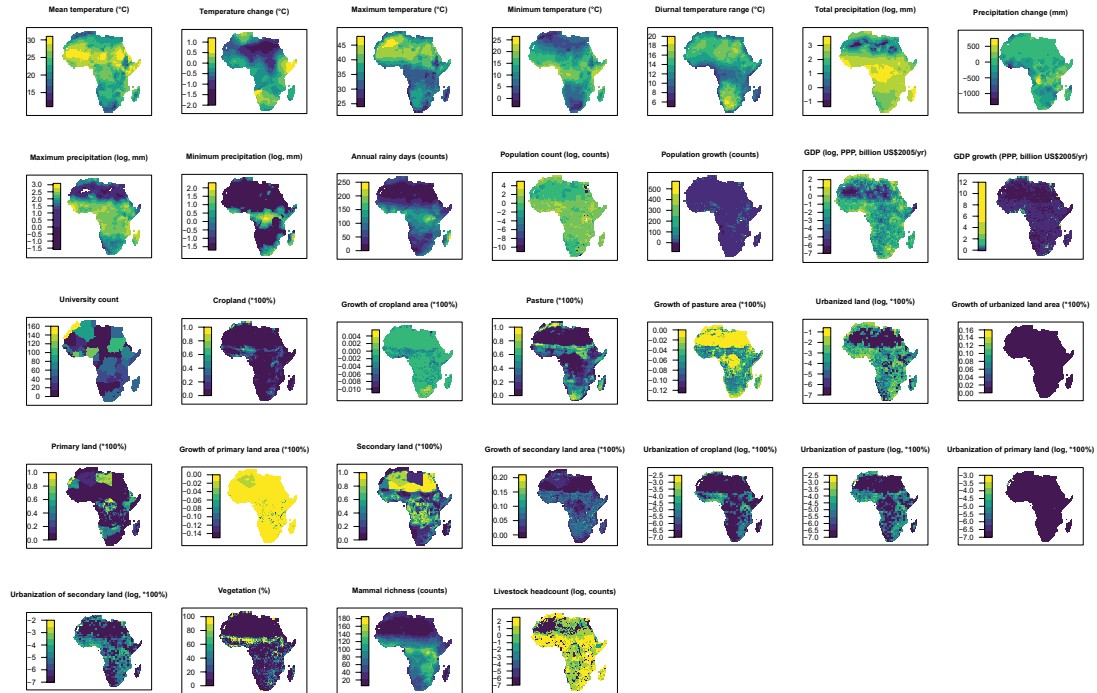

**Appendix 3—figure 11.** Distribution maps for 32 predictors in 2015 in Africa. The values of these explanatory variables and latitude in each grid cell were used to predict the virus discovery in the corresponding grid cell in Africa in 2010–2019. Explanatory variables were log transformed where necessary to get better visualization, not meaning they entered the model by logged values.

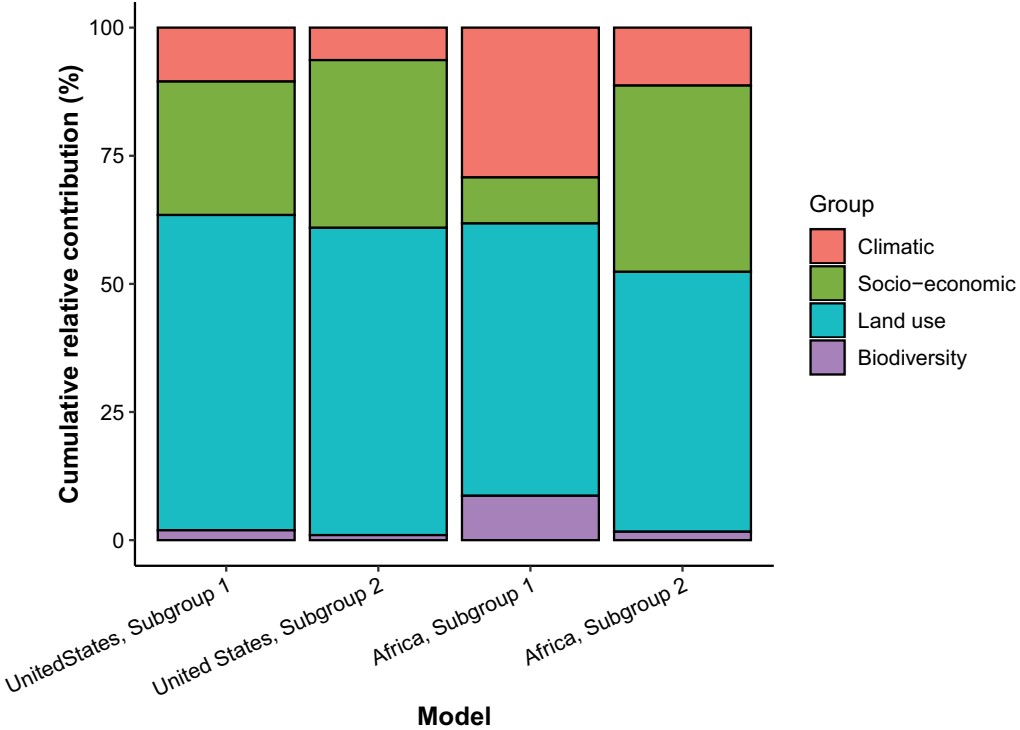

**Appendix 3—figure 12.** Cumulative relative contribution of predictors to human-infective RNA virus discovery by group in each model of subgroups. Subgroup 1 represents viruses firstly discovered from the region (United States or Africa); Subgroup 2 represents viruses firstly discovered elsewhere in the world. In the United States, virus count of Subgroup 1 and Subgroup 2 were 52 and 43, respectively. In Africa, virus count of Subgroup 1 and Subgroup 2 were 39 and 68, respectively. The relative contributions of all explanatory factors sum to 100% in each model, and each colour represents the cumulative relative contribution of all explana*tory factors within each group.*

## Appendix 4

As covariates may vary within a decade and their effects on virus discovery were likely not immediate, we performed two further sensitivity analyses by (i) matching virus discovery data and time-varying covariate data by year and (ii) testing for lag effects by matching virus discovery at year t and predictors at t-1 to t-5 year. We collected yearly data for climatic variables and land use from the same sources used in the main analysis. Yearly population data at grid level before 1970 and GDP data before 1980 are not available, so we extrapolated them back to 1901 using the yearly growth rate at country level (Source: Our World in Data). For population, the WorldPop Project provides yearly gridded data for 2000-2020 (https://www.arcgis.com/home/item.html?id=56eb0f050c614347 82f008a08331d23a), and we used the growth rate by grid to extrapolate values after 2000.

