## [Editor Report]

This study will be of interest to readers in the field of virus discovery. This study attempts to identify predictors of human-infective RNA virus discovery and predict high risk areas in a recent period in the United States, China, and Africa using an ecological modeling framework. The study has potential to inform future discovery efforts for human-infective viruses.

---

## [Decision Letter]

**Decision letter after peer review:**

Thank you for submitting your article "Predictors of human-infective RNA virus discovery in the United States, China and Africa, an ecological study" for consideration by *eLife*. Your article has been reviewed by 2 peer reviewers, and the evaluation has been overseen by a Reviewing Editor and George Perry as the Senior Editor. The following individual involved in review of your submission has agreed to reveal their identity: Benn Sartorius (Reviewer #1).

Essential revisions:

1) It seems unlikely that virus discovery effort is homogenous over the large geographic areas considered in this analysis and we would like to see additional work to address remaining discovery bias by extending the analysis to include adjustment for more granular measures of discovery effort. While we don't want to be too prescriptive about how this should be done, we would suggest that bibliographic data potentially provides a valuable and readily-available source of information about how research effort related to virus discovery varies with space and time. It should be possible to use such data to derive sub-national indicators that are both more granular than national economic indicators, and that might be expected to be more directly related to virus discovery effort. We suggest that governance indicators (transparency) and laboratory infrastructure/technology indicators are also considered as predictors.

2) Clarification is needed about how space and time is currently dealt with in the boosted regression tree model.

3) Currently it seems that virus discovery data were matched to covariate data using the nearest decade. Since there might be important changes in virus discovery effort and other covariates at finer timescales, we feel it is important to at least perform a sensitivity analysis to check for this variation and consider how it might impact on the results.

4) We would like to see a consideration of lag periods between the covariates/predictors and virus discovery. i.e. changes in predictor in t-1 year may be more predictive of virus discovery at year t.

5) Clarification is needed regarding handling of how potential collinearity between the predictors.

6) We would like the authors address the likely differences in underlying drivers/predictors for vector-borne (V) vs non-vector-borne (N) viruses and strictly zoonotic (Z) vs human transmissible (T) viruses. This could be done, for example, by using separate models.

7) It would be useful to provide an appraisal of the impact related past studies have had on improving surveillance and control.

*Reviewer #1 (Recommendations for the authors):*

This study attempts to identify predictors of human-infective RNA virus discovery and predict high risk areas in a recent period in the United States, China and Africa using a ecological modelling framework. The study is relevant in the current context and identification of areas at threat of emerging viral pathogens.

According to findings from their previous study published in 2020, the main predictors for virus discovery at the global scale were GDP-related i.e. and they concluded that this may largely have driven by research effort rather than the underlying biology. In the current study, they have attempted to focus on more restricted and homogenous regions where they suspect research effort is less heterogeneous to an attempt to identify predictors more associated with virus biology. I have some comments and concerns below that would need to be addressed in my opinion prior to a final decision being taken.

While I do understand the rationale for the restricted analysis, I am still concerned that inherent "discovery" bias in the data and how these vary by region and within country/region and across time may still drive the observed associations rather than real distal predictors of virus discovery.

Furthermore the relative lack of accurate geolocating of data in China relative to the other two regions may also misalign the covariate values attached to these data points and potential skew association particular of the more influential covariates vary substantial across space.

Was the model applied at location-year level i.e. spatial and temporal? This was not entirely clear to me under Boosted regression tree model description.

If I understood correctly virus discovery data were matched to covariate data using the nearest decade? While some covariates are likely to change more slowly, some may be far more dynamic (especially at the edge of human-animal-environmental interaction) and vary within a decade. Where any sensitivity analyses conducted to check for this variation within decade and how it might impact on the associative coefficients/influence?

Did the authors consider or test for various lag periods between the covariates/predictors and virus discovery? i.e. changes in predictor in t-1 year may be more predictive of virus discovery at year t.

Did you consider collinearity between the 33 predictors as many of these would likely be highly correlated?

What other predictors were considered in the ecological framework i.e. governance indicators (transparency) and laboratory infrastructure/technology indicators? These would seem to be important re: potential bias in reporting of newly discovered viruses for the former ("effective surveillance is challenging in less developed regions such as large parts of Africa given resource constraints" as alluded to in the introduction) and likelihood of detection for the latter ("better resources to discover new viruses" and "effective surveillance is challenging in less developed regions such as large parts of Africa given resource constraints" as alluded to in the introduction).

Did you consider separate models (#4) for vector-borne (V) vs non-vector-borne (N) viruses and strictly zoonotic (Z) vs human transmissible (T) given the likely difference (and ecological association) in underlying drivers/predictors across these envelopes?

*Reviewer #2 (Recommendations for the authors):*

As this type of work has been ongoing for some time, It would be useful to provide an appraisal of the impact these past studies have had on improving surveillance and control.

[Editors’ note: further revisions were suggested prior to acceptance, as described below.]

Thank you for resubmitting your work entitled "Predictors of human-infective RNA virus discovery in the United States, China and Africa, an ecological study" for further consideration by *eLife*. Your revised article has been evaluated by George Perry (Senior Editor) and a Reviewing Editor.

The authors have gone some way to addressing the essential revisions from the first round of reviewers and have conducted some of the additional analyses requested. However, this additional analysis does not appear to have been included in the revised manuscript files. While additional analysis is reported in the document with the responses to reviewer comments, it is important that the changes are also included in the manuscript.

Considering the requested essential revisions in turn:

Essential revision 1 asked for additional work to address discovery bias by extending the analysis to include adjustments for more granular measures of discovery effort. The response to this was to clarify aspects of the analysis (e.g. the granularity of GDP indicators) and to add some text to the discussion (lines 408-412) referring to approaches to this problem taken in other papers. There was also a short exploration (figure R1 in the response document, but not appearing in the manuscript) showing the relationship between "published human-infective RNA virus count and the total number of papers from the journals which published all human-infective RNA viruses in Web of Science", and accompanying text that argued that attempts to adjust for discovery effort used by other authors would not be applicable here. I was unable to follow this argument, and given that no additional analysis addressing discovery bias appears in the main manuscript it seems that the essential revision was not undertaken. While the changes the authors have made to the manuscript are welcome, given that (following consultation) round 1 reviewers highlighted this as the key point to address it is important that the authors reconsider how they can change the manuscript to address this point. This may just amount to additional analysis in the supplementary material demonstrating that approaches used by other authors have little applicability here as claimed in the rebuttal letter, but if so the reasoning needs to be expressed more clearly and the evidence for any such assertions clearly laid out.

Essential revision 2 appears to have been addressed adequately.

Essential revision 3 asked for sensitivity analysis to consider matching covariates at finer timescales. This additional analysis appears to have been done but has not yet (as far as I could tell) been included in the revised manuscript.

Essential revision 4 asked for additional analysis that included consideration of lag periods between the covariates/predictors and virus discovery. Again, this appears to have been done and is shown in the rebuttal letter, but I could not locate the new analysis in the revised manuscript files.

Essential revision 5 appears to have been addressed adequately.

Essential revision 6 again appears to have been addressed with additional work but this does not appear to have been included in the revised manuscript.

Essential revision 7 appears to have been addressed adequately (though note that the line numbers for the changes given in the rebuttal letter (311-330) do not correspond to areas of highlighted changes in the manuscript with changes marked).

---

## [Author Response]

Essential revisions:1) It seems unlikely that virus discovery effort is homogenous over the large geographic areas considered in this analysis and we would like to see additional work to address remaining discovery bias by extending the analysis to include adjustment for more granular measures of discovery effort. While we don't want to be too prescriptive about how this should be done, we would suggest that bibliographic data potentially provides a valuable and readily-available source of information about how research effort related to virus discovery varies with space and time. It should be possible to use such data to derive sub-national indicators that are both more granular than national economic indicators, and that might be expected to be more directly related to virus discovery effort. We suggest that governance indicators (transparency) and laboratory infrastructure/technology indicators are also considered as predictors.

Thanks for your useful suggestions. We agree the virus discovery effort is non-homogenous over the three regions included in this analysis and we have tried a couple of ways to adjust for it. The first direct method is that we restricted the study area to three specific regions where the research effort is less variable than the whole globe. Second, we included GDP, GDP growth, and university count to adjust for the discovery effort in each region as these variables partially explain the infrastructure and technology that are available for virus research. One thing we want to clarify is that the economic indicators (i.e., GDP and GDP growth) we used were all time-varying and at the grid cell level (rather than national) (see Appendix figure 6–Appendix figure 8)—they were also granular data.

We considered using bibliographic data to adjust for discovery effort at first, but we rejected this strategy after a couple of tests. Jones et al[1] estimated the discovery effort for emerging infectious diseases (EID) by calculating the number of papers published by each country (denoted by the address for every author) in the Journal of Infectious Diseases (JID) since 1973. The hypothesis is that countries publishing more papers in JID are likely to discover more EID events. We tested if this method worked for our analysis by plotting the relationship between published human-infective RNA virus count and total number of papers from all journals which published on human-infective RNA viruses in Web of Science (as of 21 Feb 2018). Both the total number of papers (Author response image 1) and total number of papers on viruses (Author response image 1) were of little relevance to the published human virus count in our database, though the number of papers had a positive relationship with the number of papers on viruses (Author response image 1). We also noted that papers in JID (highlighted in blue in Author response image 1) may not be able to fully explain the discovery efforts for newly discovered viruses. Allen et al[2] derived a different index for discovery bias, based on the spatial distribution of place names in peer-reviewed biomedical literature. The disadvantage of this method is that it may not represent the discovery effort, as many place names are not related to zoonotic viruses.

**Author response image 1. sa2fig1:** Relationship between published human-infective RNA virus count and total number of papers from the journals which published all human-infective RNA viruses in Web of Science. A, total number of papers vs. published human virus count; B, total number of papers on viruses vs. published human virus count; C, total number of papers vs. total number of papers on viruses; D, Percent of papers on viruses in each journal. (J) Infect Dis (JID) is highlighted in blue.

We were unable to include government indicators (e.g., transparency) and laboratory infrastructure/technology indicators in our model because the data is scarce. Please see our responses to comment (7) from # Reviewer 1.

Refs:

1] Jones, K. E., Patel, N. G., Levy, M. A., Storeygard, A., Balk, D., Gittleman, J. L., and Daszak, P. (2008). Global trends in emerging infectious diseases. Nature, 451(7181), 990-993.

2] Allen, T., Murray, K. A., Zambrana-Torrelio, C., Morse, S. S., Rondinini, C., Di Marco, M.,... Daszak, P. (2017). Global hotspots and correlates of emerging zoonotic diseases. Nat Commun, 8(1), 1124.

2) Clarification is needed about how space and time is currently dealt with in the boosted regression tree model.

Please see our responses to comment (3) from # Reviewer 1.

3) Currently it seems that virus discovery data were matched to covariate data using the nearest decade. Since there might be important changes in virus discovery effort and other covariates at finer timescales, we feel it is important to at least perform a sensitivity analysis to check for this variation and consider how it might impact on the results.

We have performed additional sensitivity analyses by considering yearly changes of covariates. Please see our responses to comment (4) from # Reviewer 1.

4) We would like to see a consideration of lag periods between the covariates/predictors and virus discovery. i.e. changes in predictor in t-1 year may be more predictive of virus discovery at year t.

Lag effects have been considered in the revised version. Please see our responses to comment (5) from # Reviewer 1.

5) Clarification is needed regarding handling of how potential collinearity between the predictors.

Please see our responses to comment (6) from # Reviewer 1.

6) We would like the authors address the likely differences in underlying drivers/predictors for vector-borne (V) vs non-vector-borne (N) viruses and strictly zoonotic (Z) vs human transmissible (T) viruses. This could be done, for example, by using separate models.

We additionally fitted 8 separate models for the United State and Africa. Please see our responses to comment (8) from # Reviewer 1.

(7) It would be useful to provide an appraisal of the impact related past studies have had on improving surveillance and control.

See details in our responses to #Reviewer 2.

Reviewer #1 (Recommendations for the authors):This study attempts to identify predictors of human-infective RNA virus discovery and predict high risk areas in a recent period in the United States, China and Africa using a ecological modelling framework. The study is relevant in the current context and identification of areas at threat of emerging viral pathogens.According to findings from their previous study published in 2020, the main predictors for virus discovery at the global scale were GDP-related i.e. and they concluded that this may largely have driven by research effort rather than the underlying biology. In the current study, they have attempted to focus on more restricted and homogenous regions where they suspect research effort is less heterogeneous to an attempt to identify predictors more associated with virus biology. I have some comments and concerns below that would need to be addressed in my opinion prior to a final decision being taken.1) While I do understand the rationale for the restricted analysis, I am still concerned that inherent "discovery" bias in the data and how these vary by region and within country/region and across time may still drive the observed associations rather than real distal predictors of virus discovery.

Thanks for the comments. We agree the virus discovery effort is non-homogenous over the three regions included in this study and we have tried a couple of ways to adjust for it. As you noted, the first direct method is that we restricted the study area to three specific regions where the research effort is less variable than the whole globe. Second, we included GDP, GDP growth, and university count to adjust for the discovery effort in each region as these variables partially explain the infrastructure and technology that are available for virus research. One thing we want to clarify is that the economic indicators (i.e., GDP and GDP growth) we used were all time-varying and at the grid cell level (see Appendix figure 6–Appendix figure 8)—they partially reflect the variation in discovery bias within country/region and across time.

Discovery bias is a common problem for data collected from published literature. We acknowledged identifying robust and comprehensive data on discovery effort is difficult, and we listed it as a limitation for our study (lines 396-400). Previous studies have tried to use the bibliographic data to correct for the discovery effort, but we rejected this strategy after a couple of tests (see our responses to editors’ comment 1 for details).

2) Furthermore the relative lack of accurate geolocating of data in China relative to the other two regions may also misalign the covariate values attached to these data points and potential skew association particular of the more influential covariates vary substantial across space.

Although the majority discovery locations in China involved polygon data at province level, the average number of grid cells per virus in China (243/80=3.03) were similar with the other two regions (United States: 273/95=2.87; Africa: 426/107=3.98) (Appendix table 2). More than half discovery locations in China were at the city/county level, but the average number of grid cells each city/county covered is no more than 2 (70/47<2) (Appendix table 2). By using the bootstrap resampling strategy that we described in the main manuscript can effectively avoid any potential skew associations. We re-wrote this in lines 122-124 in the revised manuscript.

3) Was the model applied at location-year level i.e. spatial and temporal? This was not entirely clear to me under Boosted regression tree model description.

Before fitting the boosted regression tree (BRT) model, we matched the virus data with all predictors by geographical coordinates and time (using the nearest decade for time-varying predictors), making sure each virus discovery event has been linked to the “correct” predictors, i.e., with the same time and location. We further explained this in line 159. However, this is different with the traditional spatiotemporal model that modelling a spatial and temporal outcome that exists at a certain time t and location x. Virus discovery is a rare event, so that if we see it as a spatial and temporal outcome there will be many zeros in the study period for a certain location. Therefore, we care more about the spatial property of the virus discovery data in this analysis, and the BRT model we were using can well correct for spatial autocorrection within the data.

4) If I understood correctly virus discovery data were matched to covariate data using the nearest decade? While some covariates are likely to change more slowly, some may be far more dynamic (especially at the edge of human-animal-environmental interaction) and vary within a decade. Where any sensitivity analyses conducted to check for this variation within decade and how it might impact on the associative coefficients/influence?

Yes, virus discovery data were matched to TIME-VARYING covariates using the nearest decade. We agree some covariates vary within a decade, but gridded covariates back to 1901 are often not available at finer timescales. In our model, covariates, particularly those are associated with the human-animal-environmental interactions including mammal species richness and livestock headcount were static. Data for climatic variables, population, GDP, and land use are time-varying and have been matched with RNA virus discovery count by decade. In the revised version, we re-collected yearly data for climatic variables and land use from the same sources used in the analysis. Yearly population data at grid level before 1970 and GDP data before 1980 are not available, and we extrapolated them back to 1901 using the yearly growth rate at country level (Source: Our World in Data). For population, the WorldPop provides yearly gridded data for 2000-2020[1], and we used the growth rate by grid to extrapolate values after 2000.

Sensitivity analyses were performed by matching virus discovery data and time-varying covariate data by YEAR. Though there are several changes of relative contribution, the leading predictors broadly keep consistent with our model based on data matched by decade (Author response image 2).

**Author response image 2. sa2fig2:** Relative contribution of predictors to human-infective RNA virus discovery in three regions. Virus discovery data were matched to time-varying covariate data by year. (A) United States. (B) China. (C) Africa. The boxplots show the median (black bar) and interquartile range (box) of the relative contribution across 1000 replicate models, with whiskers indicating minimum and maximum and black dots indicating outliers.

Data sources

1] https://www.arcgis.com/home/item.html?id=56eb0f050c61434782f008a08331d23a

5) Did the authors consider or test for various lag periods between the covariates/predictors and virus discovery? i.e. changes in predictor in t-1 year may be more predictive of virus discovery at year t.

Thanks for the useful suggestion. We have tested for 1~5-year lag between the predictors and virus discovery, and the relative contribution of most predictors stay consistent with the model based on data matched by 0-year lag (e.g., results of 1-year lag shown in Author response image 3).

**Author response image 3. sa2fig3:** Relative contribution of predictors to human-infective RNA virus discovery in three regions. Virus discovery data at year t were matched to time-varying covariate data at year t-1. (A) United States. (B) China. (C) Africa. The boxplots show the median (black bar) and interquartile range (box) of the relative contribution across 1000 replicate models, with whiskers indicating minimum and maximum and black dots indicating outliers.

6) Did you consider collinearity between the 33 predictors as many of these would likely be highly correlated?

We do not consider collinearity between the 33 predictors as the BRT model we used can automatically capture complex relationships and interactions between variables (lines 146-148) and provides robust estimations even if collinearity exist. In spite of this, we took the United States as the example and validated our results by removing predictors with high correlation (correlation coefficient > 0.7 in Author response image 4), and the relative contributions of the remaining predictors (Author response image 5) stay consistent with our model in the main manuscript.

**Author response image 4. sa2fig4:** Correlation matrix for predictors. Positive correlations are displayed in blue and negative correlations in red colour. Spearman’s rank correlation test was used. Colour intensity is proportional to the correlation coefficients.

**Author response image 5. sa2fig5:** Relative contribution of predictors to human-infective RNA virus discovery in the United States by removing high-correlated predictors. The boxplots show the median (black bar) and interquartile range (box) of the relative contribution across 1000 replicate models, with whiskers indicating minimum and maximum and black dots indicating outliers.

7) What other predictors were considered in the ecological framework i.e. governance indicators (transparency) and laboratory infrastructure/technology indicators? These would seem to be important re: potential bias in reporting of newly discovered viruses for the former ("effective surveillance is challenging in less developd regions such as large parts of Africa given resource con”traints" as alluded to in the introduction) and likelihood of detection for the “atter ("better resources to discover new”virus“s" and "effective surveillance is challenging in less developed regions such as large parts of Africa given resource con”traints" as alluded to in the introduction).

We agree that government indicators (e.g., transparency) could partially correct for the potential bias in reporting and the laboratory infrastructure/technology indicators for the potential bias in detecting novel viruses. However, we did not include them in our model as we cannot obtain the appropriate data. Some agencies such as the World Bank[1] and Our World in Data[2] provide data on national government indicators, but it seems none of them provide data at the sub-national level. There are some sub-national data on the quality of governance for Europe[3], but there is not much for regions of interest in our study. Data on laboratory infrastructure/technology indicators are also scarce. There is some data on biosafety level 4 (BSL4) labs around the world[4], but of the 42 labs where foundation dates are available, approximately half have been established in the recent decade, making less contribution to virus discoveries in earlier years. In our study, we included the university count in the model, and should partially adjust for the university labs (Quite a few viruses in our study have been discovered in university labs). We acknowledged identifying robust and comprehensive data on virus discovery effort is difficult, and we re-wrote this as a limitation for our study (lines 396-400).

Data links

1] https://info.worldbank.org/governance/wgi/

2] https://ourworldindata.org/grapher/government-transparency-index

3] https://citeseerx.ist.psu.edu/viewdoc/download?doi=10.1.1.367.6086&rep=rep1&type=pdf

4] Mapping+BSL4+Labs+Globally+EMBARGOED+until+27+May+2021+1800+CET.pdf

8) Did you consider separate models (#4) for vector-borne (V) vs non-vector-borne (N) viruses and strictly zoonotic (Z) vs human transmissible (T) given the likely difference (and ecological association) in underlying drivers/predictors across these envelopes?

Thanks for your suggestion. In our global analysis (Zhang et al. 2020), we have fitted separate models for these four categories of viruses and have detected the differences in underlying predictors across them, i.e., vector-borne viruses and strictly zoonotic viruses are more associated with climate and biodiversity whereas non-vector-borne viruses and human transmissible viruses are more associated with GDP and urbanization. In this study, we additionally fitted 8 separate models for the United States and Africa (we were unable to perform subgroup analysis for China given the small sample size in some subgroups, lines 213-216) and found that the pattern of differences detected from our global analysis holds true in Africa (Author response images 6-7, image 10). In the Unites States, we did not see the similar pattern (Author response images 8-10) and this is likely due to the smaller proportions of vector-borne and strictly zoonotic viruses in this region (lines 213-216).

**Author response image 6. sa2fig6:** Relative contribution of explanatory factors to human RNA virus discovery in the stratified model by transmissibility in Africa. (A) Strictly zoonotic, (B) Transmissible in humans. The boxplots show the median (black bar) and interquartile range (box) of the relative contribution across 1000 replicate models, with whiskers indicating minimum and maximum and black dots indicating outliers.

**Author response image 7. sa2fig7:** Relative contribution of explanatory factors to human RNA virus discovery in the stratified model by transmission mode in Africa. (A) Vector-borne, (B) Non-vector- borne. The boxplots show the median (black bar) and interquartile range (box) of the relative contribution across 1000 replicate models, with whiskers indicating minimum and maximum and black dots indicating outliers.

**Author response image 8. sa2fig8:** Relative contribution of explanatory factors to human RNA virus discovery in the stratified model by transmissibility in the United States. (A) Strictly zoonotic, (B) Transmissible in humans. The boxplots show the median (black bar) and interquartile range (box) of the relative contribution across 1000 replicate models, with whiskers indicating minimum and maximum and black dots indicating outliers..

**Author response image 9. sa2fig9:** Relative contribution of explanatory factors to human RNA virus discovery in the stratified model by transmission mode in the United States. (A) Vector-borne, (B) Non-vector- borne. The boxplots show the median (black bar) and interquartile range (box) of the relative contribution across 1000 replicate models, with whiskers indicating minimum and maximum and black dots indicating outliers.

**Author response image 10. sa2fig10:** Cumulative relative contribution of predictors to human-infective RNA virus discovery by group in each model of different regions. The relative contributions of all explanatory factors sum to 100% in each model, and each colour represents the cumulative relative contribution of all explanatory factors within each group.

Reviewer #2 (Recommendations for the authors):As this type of work has been ongoing for some time, It would be useful to provide an appraisal of the impact these past studies have had on improving surveillance and control.

Thanks for your suggestions and we have provided an appraisal in lines 311-330 of the impact these past studies have had on improving surveillance and control.

[Editors’ note: further revisions were suggested prior to acceptance, as described below.]

The authors have gone some way to addressing the essential revisions from the first round of reviewers and have conducted some of the additional analyses requested. However, this additional analysis does not appear to have been included in either of the revised manuscript files (neither the clean version nor the version with changes marked which doesn't include figures or appendices). While additional analysis is reported in the document with the responses to reviewer comments, it is important that the changes are also included in the manuscript.Considering the requested essential revisions in turn:Essential revision 1 asked for additional work to address discovery bias by extending the analysis to include adjustments for more granular measures of discovery effort. The response to this was to clarify aspects of the analysis (e.g. the granularity of GDP indicators) and to add some text to the discussion (lines 408-412) referring to approaches to this problem taken in other papers. There was also a short exploration (figure R1 in the response document, but not appearing in the manuscript) showing the relationship between "published human-infective RNA virus count and the total number of papers from the journals which published all human-infective RNA viruses in Web of Science", and accompanying text that argued that attempts to adjust for discovery effort used by other authors would not be applicable here. I was unable to follow this argument, and given that no additional analysis addressing discovery bias appears in the main manuscript it seems that the essential revision was not undertaken. While the changes the authors have made to the manuscript are welcome, given that (following consultation) round 1 reviewers highlighted this as the key point to address it is important that the authors reconsider how they can change the manuscript to address this point. This may just amount to additional analysis in the supplementary material demonstrating that approaches used by other authors have little applicability here as claimed in the rebuttal letter, but if so the reasoning needs to be expressed more clearly and the evidence for any such assertions clearly laid out.

Thanks for your suggestions. We have now added in lines 139-145 (Methods) and lines 419-421 (Discussion) in the manuscript that we test approaches used by other authors for adjusting for discovery effort but found they were not appliable for our data. Details for how we test these approaches are now put in Appendix 2. These approaches used by other authors hypothesized that there is a linear relationship between the frequency of published papers and published novel pathogens in scientific journals and so authors’ addresses in these scientific journals could be used to adjust for spatial discovery effort. However, according to our test, virus-related journals publishing more papers are not necessarily publishing more novel human RNA viruses.

Essential revision 2 appears to have been addressed adequately.

Thanks for the comment.

Essential revision 3 asked for sensitivity analysis to consider matching covariates at finer timescales. This additional analysis appears to have been done but has not yet (as far as I could tell) been included in the revised manuscript.

Thanks for the comment. We only mentioned in the text we did sensitivity analysis by matching covariates and virus discovery data by year in lines 172-173 and reported the results in 273-277 in our last revised manuscript. Now we have additionally presented the result in Appendix 3—figure 7.

Essential revision 4 asked for additional analysis that included consideration of lag periods between the covariates/predictors and virus discovery. Again, this appears to have been done and is shown in the rebuttal letter, but I could not locate the new analysis in the revised manuscript files.

Thanks for the comment. We only mentioned in the text we did sensitivity analysis by matching virus discovery at year t and predictors at t-1 to t-5 year in lines 173-174 and reported the results in 273-277 in our last revised manuscript. Now we have additionally presented the result in Appendix 3—figure 8.

Essential revision 5 appears to have been addressed adequately.

Thanks for the comment.

Essential revision 6 again appears to have been addressed with additional work but this does not appear to have been included in the revised manuscript.

We did not include subgroup analyses by transmission modes in our last revised manuscript because (i) we could not do the analysis for China given the small sample size in some subgroups and (ii) Again due to the small sample size for vector-borne viruses and strictly zoonotic viruses in this United States, it gives inconsistent results with Africa. We do not want to address these inconclusive findings in the main manuscript.

Essential revision 7 appears to have been addressed adequately (though note that the line numbers for the changes given in the rebuttal letter (311-330) do not correspond to areas of highlighted changes in the manuscript with changes marked).

Thanks for the comment. Sorry we gave the wrong line numbers—we were asked to move the funding information into the abstract after we submitted the revised version, and this has made all line numbers changed. We forgot to correct these in the response letter when we re-submitted.